# SVRG and Beyond via Posterior Correction

**Nico Daheim** [1]   **Thomas Möllenhoff** [2]   **Ming Liang Ang** [3]   **Mohammad Emtiyaz Khan** [2]

## Abstract

Stochastic Variance Reduced Gradient (SVRG) and its variants aim to speed-up training by using gradient corrections. Originally proposed over a decade ago, these methods have never been connected to any Bayesian method at a fundamental level. Here, we fill this gap and derive surprising new connections of SVRG to a recently proposed Bayesian method called 'posterior correction'. Our main contribution is to show that SVRG can be recovered as a special case of posterior-correction over isotropic-Gaussian posteriors. Novel extensions of SVRG are automatically obtained by using more flexible exponential-family posteriors. We derive two new such extensions by using Gaussian families: a Newton-like variant with novel Hessian corrections, and an Adam-like extension that scales to large problems. Our work is the first to connect SVRG to Bayes and use it to speed-up training.

## 1. Introduction

Variance reduction is a powerful technique to speed-up stochastic optimization. For example, stochastic variance reduced gradient (SVRG) uses full-batch gradients to stabilize future mini-batch updates (Johnson & Zhang, 2013). The method originates in the works of Roux et al. (2012); Shalev-Shwartz & Zhang (2013); Schmidt et al. (2017) which required occasional full-batch gradient computations. Since then, a large number of variants have been proposed exploring various aspects of this method (Nguyen et al., 2017; Fang et al., 2018; Cutkosky & Orabona, 2019).

Despite more than a decade of work on these methods, they

---

[1] Ubiquitous Knowledge Processing Lab (UKP Lab), Department of Computer Science, Technical University of Darmstadt and National Research Center for Applied Cybersecurity ATHENE, Germany [2] RIKEN Center for Advanced Intelligence Project, Tokyo, Japan [3] University College London. Correspondence to: Nico Daheim <nico.daheim@tu-darmstadt.de>, Mohammad Emtiyaz Khan <emtiyaz.khan@riken.jp>.

*Proceedings of the 43rd International Conference on Machine Learning*, Seoul, South Korea. PMLR 306, 2026. Copyright 2026 by the author(s).

have never been connected to any Bayesian method. This is not because variance reduction is not useful for Bayesian methods. In fact, some works have attempted to use it to speed up Bayesian procedures (Mandt & Blei, 2014; Zhang et al., 2019). However, a deeper and more fundamental connection does not exist.

In this paper, we fill this gap and show a previously unknown connection between SVRG and Bayes. Our first contribution is to show that SVRG can be derived as a special case of a recently proposed Bayesian method called posterior-correction (PoCo) (Khan, 2025); see Fig. 1. The new connection is surprising because PoCo is a unifying mechanism for knowledge adaptation methods such as continual learning and model merging, and is not directly related to variance-reduction. Our result provides the first direct connection between such adaptation methods and variance reduction. It offers a new perspective where gradient-corrections in SVRG can be seen as a mechanism of knowledge-transfer between old and new gradients.

Our second contribution is to derive new extensions which cannot easily be derived with existing SVRG techniques. For instance, we derive a new SVRG variant that uses Newton steps where, in addition to gradient corrections, Hessian corrections are also employed. This differs fundamentally from most works on Newton-style SVRG that only use corrections for the gradient and never for the Hessian (Derezinski, 2025; Sadiev et al., 2024; Garg et al., 2025; Sun et al., 2025). In contrast, our new variant automatically emerges when posterior correction is applied using full-Gaussian posteriors. We stress that this does not happen when SVRG is naively applied to Bayesian algorithms. We show that the 'natural-gradients' used in Alg. 2 play an important role to go beyond SVRG.

Finally, we derive an Adam-like extension that scales well to large problems such as those in deep learning. This variant is obtained by using Gaussians with diagonal covariances which leads to a variance-reduced version of the recently-proposed Improved Variational Online Newton (IVON) optimizer (Shen et al., 2024). We call the new variant IVON-PoCo and show that, on convex problems, it accelerates IVON's training in a similar way as SVRG accelerates SGD (Fig. 2). A similar improvement is observed for training GPT-2 from scratch, although no ef-

**Algorithm 1** SVRG

1: Initialize $\boldsymbol{\theta}_{\text{in}}$
2: **while** not converged **do**
3: $\quad\mathbf{g}_{\text{out}} \leftarrow \sum_{i=1}^{N} \nabla\ell_i(\boldsymbol{\theta}_{\text{in}})$
4: $\quad\boldsymbol{\theta}_{\text{out}} \leftarrow \boldsymbol{\theta}_{\text{in}}$
5: $\quad$**for** $t = 1, 2, \ldots, m$ **do**
6: $\qquad$Randomly pick $i \in \{1, 2, \ldots, N\}$
7: $\qquad\mathbf{g}_{\text{in}} \leftarrow \nabla\ell_i(\boldsymbol{\theta}_{\text{in}}) - \nabla\ell_i(\boldsymbol{\theta}_{\text{out}}) + \frac{1}{N}\mathbf{g}_{\text{out}}$
8: $\qquad\boldsymbol{\theta}_{\text{in}} \leftarrow \boldsymbol{\theta}_{\text{in}} - \eta\mathbf{g}_{\text{in}}$
9: $\quad$**end for**
10: **end while**

**Algorithm 2** Posterior Correction (PoCo)

1: Initialize $\boldsymbol{\lambda}_{\text{in}}$
2: **while** not converged **do**
3: $\quad\widetilde{\mathbf{g}}_{\text{out}} \leftarrow \sum_{i=1}^{N} \widetilde{\nabla}\mathcal{L}_i(\boldsymbol{\lambda}_{\text{in}})$
4: $\quad\boldsymbol{\lambda}_{\text{out}} \leftarrow \boldsymbol{\lambda}_{\text{in}}$
5: $\quad$**for** $t = 1, 2, \ldots, m$ **do**
6: $\qquad$Randomly pick $i \in \{1, 2, \ldots, N\}$
7: $\qquad\widetilde{\mathbf{g}}_{\text{in}} \leftarrow \widetilde{\nabla}\mathcal{L}_i(\boldsymbol{\lambda}_{\text{in}}) - \widetilde{\nabla}\mathcal{L}_i(\boldsymbol{\lambda}_{\text{out}}) + \frac{1}{N}\widetilde{\mathbf{g}}_{\text{out}}$
8: $\qquad\boldsymbol{\lambda}_{\text{in}} \leftarrow (1-\eta)\boldsymbol{\lambda}_{\text{in}} - \eta N\widetilde{\mathbf{g}}_{\text{in}}$
9: $\quad$**end for**
10: **end while**

*Figure 1.* Pseudo-code for SVRG (left) and our Bayesian generalization (right). The latter replaces all instances of parameter $\boldsymbol{\theta}$ and loss-gradients $\nabla\ell_i$ in SVRG by the natural parameter $\boldsymbol{\lambda}$ of the posterior $q(\boldsymbol{\theta})$ and natural gradient $\widetilde{\nabla}$ of the expected loss $\mathcal{L}_i = \mathbb{E}_q[\ell_i]$, respectively. Two other differences in the right pseudo-code are shown in red. Our main result is to show that if we set $q = \mathcal{N}(\mathbf{m}, \mathbf{I})$, an isotropic Gaussian, then SVRG can be derived as a special case of PoCo where $\boldsymbol{\theta}$ is replaced by $\mathbf{m}$; see Thm. 2 and Alg. 3.

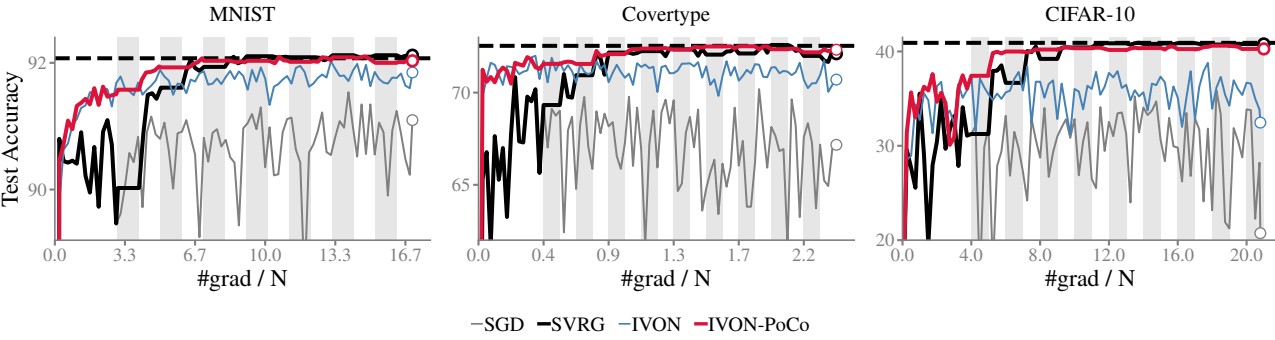

*Figure 2.* We propose IVON-PoCo to accelerate IVON in a similar way as SVRG accelerates SGD. These results are for logistic regression on three datasets (MNIST, Covertype, and CIFAR-10). We show accuracy vs. the number of gradient evaluations per data example. The horizontal dashed-gray line is the performance obtained by a full-batch L-BFGS run. Gray vertical bars indicate the computation of full-batch gradients in SVRG and IVON-PoCo. After such computations, we often observe a jump in the performance.

fective gains are observed in runtime. This result matches those of Defazio & Bottou (2019) who also showed the ineffectiveness of variance reduction on deep learning problems but for smaller models than ours. We provide many such experiments on LLM and image classifier training with a hope that our work can be useful in the near future to improve the effectiveness of variance reduction for deep learning. Code is available publicly on GitHub.

## 2. SVRG and Bayesian Methods

SVRG aims to improve stochastic optimization by reducing gradient variance. For illustration, let us consider the following Empirical Risk Minimization (ERM) problem:

$$\boldsymbol{\theta}_* = \arg\min_{\boldsymbol{\theta}\in\Theta} \sum_{i=0}^{N} \ell_i(\boldsymbol{\theta}), \tag{1}$$

where $\ell_i$ is the loss for examples $i = 1, 2, \ldots, N$ and $\ell_0$ is a regularizer. To get rid of the regularizer, it is common in the

SVRG literature to redefine the loss $\ell_i$ as $\ell_i + \ell_0/N$ to absorb the regularizer, and consider minimizing $\sum_{i=1}^{N} \ell_i(\boldsymbol{\theta})$. This way we only have to sample $\ell_i$ and not worry about $\ell_0$. We will follow the same convention.

To minimize such objectives, SGD employs the following update using a randomly drawn example $i$,

$$\boldsymbol{\theta} \leftarrow \boldsymbol{\theta} - \eta\nabla\ell_i(\boldsymbol{\theta}). \tag{2}$$

The gradient $\nabla\ell_i(\boldsymbol{\theta})$ has a larger variance compared to the full-batch gradient which can lead to instability. SVRG aims to reduce this variance by using a double-loop algorithm shown in Alg. 1. There, full-batch gradients are computed in the outer loop by using an older parameter $\boldsymbol{\theta}_{\text{out}}$, while we update another parameter $\boldsymbol{\theta}_{\text{in}}$ in the inner loop as

$$\boldsymbol{\theta}_{\text{in}} \leftarrow \boldsymbol{\theta}_{\text{in}} - \eta\Big[\nabla\ell_i(\boldsymbol{\theta}_{\text{in}}) - \nabla\ell_i(\boldsymbol{\theta}_{\text{out}}) + \frac{1}{N}\sum_{j=1}^{N}\nabla\ell_j(\boldsymbol{\theta}_{\text{out}})\Big]. \tag{3}$$

This update is done for a few steps in the inner loop by using a constant learning rate $\eta$. The inner steps can be seen as using fresh new $\nabla \ell_i(\boldsymbol{\theta}_{\text{in}})$ to swap out the older $\nabla \ell_i(\boldsymbol{\theta}_{\text{out}})$ from the full-batch gradient. Essentially, we are 'correcting' the gradients to stop them from going stale.

SVRG is built upon earlier ideas in Stochastic Average Gradient (SAG) (Roux et al., 2012; Schmidt et al., 2017) and Stochastic Dual Coordinate Descent (SDCA) (Shalev-Shwartz & Zhang, 2013), where the gradients of all examples are stored separately and these entries are refreshed whenever corresponding examples are chosen during updating. Over the years, many new practical variants have been presented, for instance, SAGA (Defazio et al., 2014), SARAH (Nguyen et al., 2017), and SPIDER (Fang et al., 2018), among many other such proposals (Babanezhad Harikandeh et al., 2015; Lei et al., 2017; Cutkosky & Orabona, 2019; Dubois-Taine et al., 2022). In practice, it is often useful to not use full-batches but rather *mega*-batches which can be, for example, 10-50 times larger than the size of the mini-batches. It also helps to use Adam and down-weight the corrections (Yin et al., 2025). Overall, many works have shown that variance reduction is useful to speed-up optimization.

### 2.1. Variational Bayes (VB)

Despite more than a decade of work on SVRG and variants, these methods have never been connected to any Bayesian method. VB generalizes ERM by replacing the optimization over $\boldsymbol{\theta}$ by one over distributions $q(\boldsymbol{\theta}) \in \mathcal{Q}$. VB was originally proposed to find a $q_*(\boldsymbol{\theta})$ that best matches the *posterior* distribution of a Bayesian model. That is, given a prior $p_0(\boldsymbol{\theta})$ and likelihoods $\tilde{p}_i(\boldsymbol{\theta})$, we want

$$q_* \approx \frac{1}{\mathcal{Z}} p_0 \prod_{i=1}^{N} \tilde{p}_i,$$

where $\mathcal{Z}$ is the normalizing constant. However, this same process can also be used to generalize ERM by setting the prior and likelihood according to the loss, for instance,

$$p_0 \propto \exp(-\ell_0) \quad \text{and} \quad \tilde{p}_i \propto \exp(-\ell_i),$$

for all $i = 1, 2, \ldots, N$. With these choices, the ERM problem in Eq. 1 translates to the search for the mode of the so-called Gibbs posterior (Zhang, 1999) defined as $\prod_{i=0}^{N} \exp(-\ell_i)/\mathcal{Z}$. The *variational* formulation (Saul et al., 1996) enables us to express the search for $q_*$ as an optimization problem which has a form similar to Eq. 1:

$$q_* = \arg\min_{q \in \mathcal{Q}} \sum_{i=1}^{N} \underbrace{\mathbb{E}_q[\ell_i]}_{=\mathcal{L}_i} + \mathbb{D}_{\text{KL}}[q \| p_0]. \tag{4}$$

The first term now involves expectation of losses under $q$, while an additional regularization is added in the second term through the Kullback-Leibler (KL) divergence. Ultimately, this formulation enables the application of VB to solve an ERM problem (Khan & Rue, 2023).

The VB objective can be optimized by using a standard gradient-based method (Ranganath et al., 2014), and SVRG can be directly applied on top of it (Zhang et al., 2019), for example, to accelerate stochastic variational inference (Mandt & Blei, 2014). These are useful for variance reduction of stochastic VB, but we wonder if there are more, deeper and fundamental, connections of SVRG to Bayes, and whether it is possible to use Bayesian principles to go beyond SVRG. To our knowledge, no previous work has addressed these questions, and we aim to fill this gap here.

### 2.2. The Bayesian Learning Rule (BLR)

When thinking of generalizations of SVRG using Bayesian principles, we are inspired by the work of Khan & Rue (2023) who unify and generalize various existing learning algorithms. They propose a new algorithm called the Bayesian learning rule (BLR) and use it to derive a wide variety of learning algorithms as special cases. To give an example, in Khan & Rue (2023, App. C), they show that the solution $\boldsymbol{\theta}_*$ of Eq. 1 can be recovered from $q_*$ in two steps. First, we fix $\mathcal{Q}$ to be the set of isotropic Gaussians $q(\boldsymbol{\theta}) = \mathcal{N}(\boldsymbol{\theta}|\mathbf{m}, \mathbf{I})$ where $\mathbf{m}$ is the mean. Second, we use the 'delta' method to approximate the expected loss

$$\mathbb{E}_q[\ell_i] \approx \ell_i(\mathbf{m}). \tag{5}$$

With these steps, an optimization over $\mathbf{m}$ reduces to Eq. 1. As a result, using an SGD over Eq. 4 with respect to $\mathbf{m}$ recovers Eq. 2 as a special case. We will use a similar procedure to derive novel extensions of SVRG.

Khan & Rue (2023) present many such examples deriving a wide variety of learning algorithms by using specific exponential-family (EF) distributions. EF distributions are defined by using a sufficient statistic $\mathbf{T}(\boldsymbol{\theta})$ and parameterized through the 'natural' parameter $\boldsymbol{\lambda}$ as shown below,

$$q(\boldsymbol{\theta}) \propto h(\boldsymbol{\theta}) \exp\left(\boldsymbol{\lambda}^\top \mathbf{T}(\boldsymbol{\theta})\right), \tag{6}$$

where $h(\boldsymbol{\theta})$ is a base probability measure. As an example, we can write isotropic Gaussians in this form by defining $\boldsymbol{\lambda} = \mathbf{m}, \mathbf{T}(\boldsymbol{\theta}) = \boldsymbol{\theta}$ and $h(\boldsymbol{\theta}) = \exp(-\boldsymbol{\theta}^\top \boldsymbol{\theta}/2)$, to write

$$\mathcal{N}(\boldsymbol{\theta}|\mathbf{m}, \mathbf{I}) \propto e^{-\frac{1}{2}\boldsymbol{\theta}^\top \boldsymbol{\theta}} \exp\left(\mathbf{m}^\top \boldsymbol{\theta}\right). \tag{7}$$

The BLR update uses natural-gradient descent (NGD) over $\boldsymbol{\lambda}$ to solve Eq. 4. In this approach, the use of natural-parameters and natural-gradients is crucial to unify learning algorithms. The natural gradient is defined by using the Fisher $\mathbf{F}(\boldsymbol{\lambda})$ but, thanks to a property of EF distribution, we never have to deal with it. The natural gradients can be

easily obtained by using 'vanilla' gradients with respect to the expectation parameters $\boldsymbol{\mu} = \mathbb{E}_q[\mathbf{T}(\boldsymbol{\theta})]$,

$$\widetilde{\nabla}\mathcal{L}_i = \mathbf{F}(\boldsymbol{\lambda})^{-1}\nabla\mathcal{L}_i = \nabla_{\boldsymbol{\mu}}\mathcal{L}_i. \qquad (8)$$

With this notation, the NGD for Eq. 4 can be written as

$$\boldsymbol{\lambda} \leftarrow \boldsymbol{\lambda} - \eta \left[ \sum_{i=1}^{N} \widetilde{\nabla}\mathcal{L}_i(\boldsymbol{\lambda}) + \widetilde{\nabla}\mathbb{D}_{\mathrm{KL}}[q \,\|\, p_0] \right], \qquad (9)$$

which can be simplified further by expanding the KL term. Specifically, if $q$ and $p_0$ are defined with the same base measure $h(\boldsymbol{\theta})$, then we can write

$$\widetilde{\nabla}\mathbb{D}_{\mathrm{KL}}[q \,\|\, p_0] = \widetilde{\nabla}\mathbb{E}_q\left[\log\frac{q}{p_0}\right] = \boldsymbol{\lambda} + \widetilde{\nabla}\mathbb{E}_q[\ell_0].$$

The second equality is due to a result by (Khan & Rue, 2023, App. B) which shows that $\widetilde{\nabla}\mathbb{E}_q[\log q] = \boldsymbol{\lambda}$. Substituting the above in Eq. 9, the NGD can be written as

$$\boldsymbol{\lambda} \leftarrow (1 - \eta)\boldsymbol{\lambda} - \eta \sum_{i=0}^{N} \widetilde{\nabla}\mathcal{L}_i(\boldsymbol{\lambda}). \qquad (10)$$

This is the BLR update in the natural-parameter space. Note that, similarly to Eq. 1, the sum includes $i = 0$ which corresponds to the regularizer term. A stochastic version can also be used in a similar fashion as Eq. 2.

The BLR update can also be written in a form similar to Bayes' rule, which we will use later to connect to posterior correction. First, we plug $\boldsymbol{\lambda}$ in Eq. 6 to write

$$e^{\boldsymbol{\lambda}^{\top}\mathbf{T}(\boldsymbol{\theta})} \leftarrow e^{(1-\eta)\boldsymbol{\lambda}^{\top}\mathbf{T}(\boldsymbol{\theta})} \prod_{i=0}^{N} \exp\left(-\eta\widetilde{\nabla}\mathcal{L}_i(\boldsymbol{\lambda})^{\top}\mathbf{T}(\boldsymbol{\theta})\right).$$

Then, we define the following function:

$$\hat{\ell}_i(\boldsymbol{\theta}) = \widetilde{\nabla}\mathcal{L}_i(\boldsymbol{\lambda})^{\top}\mathbf{T}(\boldsymbol{\theta}). \qquad (11)$$

Using this, we can rewrite the BLR as an update of $q$,

$$q \leftarrow q^{1-\eta} \prod_{i=0}^{N} \exp\left(-\eta\hat{\ell}_i\right). \qquad (12)$$

The update is equivalent to Bayesian inference in a model where the prior is $q^{1-\eta}\exp(-\hat{\ell}_0)^{\eta}$ and likelihoods are obtained by using the sites $\hat{\ell}_i$ which are essentially linear surrogates of $\ell_i$ (linear with respect to $\mathbf{T}(\boldsymbol{\theta})$). The site functions are commonly employed in the Bayesian literature in the context of expectation propagation (Minka, 2001). The BLR defines an equivalent for VB (Khan & Nielsen, 2018).

# 3. Beyond SVRG via Posterior Correction

We will now show that a recently-proposed Bayesian approach called Posterior Correction (PoCo) (Khan, 2025) generalizes SVRG. Afterwards, we exploit this connection to derive novel methods beyond SVRG.

## 3.1. Posterior Correction

Posterior Correction is a technique for knowledge transfer and adaptation for tasks such as continual learning and model merging. The main idea is to use previously computed posteriors and adapt them to new situations. A variety of strategies are proposed by Khan (2025) but, in this paper, we will focus on a variant proposed in Khan (2025, Sec. 4.2) to speed-up variational training. In this variant, an older $\boldsymbol{\lambda}_{\mathrm{out}}$ is used to build the following posterior,

$$\hat{q}_{\mathrm{out}} \leftarrow \prod_{i=0}^{N} \exp\left(-\hat{\ell}_{i|\mathrm{out}}\right),$$

$$\text{where } \hat{\ell}_{i|\mathrm{out}}(\boldsymbol{\theta}) = \widetilde{\nabla}\mathcal{L}_i(\boldsymbol{\lambda}_{\mathrm{out}})^{\top}\mathbf{T}(\boldsymbol{\theta}). \qquad (13)$$

This posterior is then used to *correct* future updates. This is done by multiplying and dividing the left and right side of Eq. 13 in the BLR given in Eq. 12,

$$q \leftarrow q^{1-\eta} \prod_{i=0}^{N} \exp\left(-\eta\hat{\ell}_i\right) \frac{\hat{q}_{\mathrm{out}}^{\eta}}{\prod_{i=0}^{N} \exp\left(-\eta\hat{\ell}_{i|\mathrm{out}}\right)}. \qquad (14)$$

Essentially, we have multiplied the BLR by 1, which does not change anything, but it allows us to rearrange the update in the following convenient form:

$$q \leftarrow q^{1-\eta}\hat{q}_{\mathrm{out}}^{\eta} \prod_{i=0}^{N} \exp\left(-\eta\left[\hat{\ell}_i - \hat{\ell}_{i|\mathrm{out}}\right]\right). \qquad (15)$$

This is posterior correction: the site functions $\hat{\ell}_i$ in the BLR update are now corrected using the older sites $\hat{\ell}_{i|\mathrm{out}}$. Khan (2025) only considered the full-batch setting and made no reference to variance reduction. We fill this gap by using a double-loop algorithm similar to SVRG.

## 3.2. Posterior Correction Generalizes SVRG

To show that SVRG can be recovered using Posterior correction, we propose a mini-batch version of Eq. 15. To do so, similarly to the SVRG case, we need to first redefine the loss $\ell_i \leftarrow (\ell_i + \ell_0/N)$ to absorb the regularizer. Then, we rename $q$ and $\hat{\ell}_i$ in Eq. 15 by $q_{\mathrm{in}}$ and $\hat{\ell}_{i|\mathrm{in}}$, respectively. After this, we can write an unbiased stochastic version of Eq. 15 where we sample just one example,

$$q_{\mathrm{in}} \leftarrow q_{\mathrm{in}}^{1-\eta}\hat{q}_{\mathrm{out}}^{\eta} \exp\left(-\eta N\left[\hat{\ell}_{i|\mathrm{in}} - \hat{\ell}_{i|\mathrm{out}}\right]\right). \qquad (16)$$

This update can be simplified to a form analogous to Eq. 3. To do so, we need to write the update in terms of $\boldsymbol{\lambda}$, as it is done for the BLR update in Eq. 10. This is obtained by collecting the $\boldsymbol{\lambda}_{\mathrm{in}}$ terms from the definitions of $q_{\mathrm{in}}$, $\hat{q}_{\mathrm{out}}$, $\hat{\ell}_{i|\mathrm{in}}$ and $\hat{\ell}_{i|\mathrm{out}}$. A detailed derivation is in App. A where we

**Algorithm 3** VSGD-PoCo: Variational SGD with Posterior Correction

**Initialize:** Number of inner steps $m$, learning rates $\eta$
1: Initialize $\mathbf{m}_{in}$.
2: **while** not converged **do**
3:     $\boldsymbol{\theta}_{in} \leftarrow \mathbf{m}_{in} + \boldsymbol{\epsilon}$ with $\boldsymbol{\epsilon} \sim \mathcal{N}(0, \mathbf{I})$
4:     $\mathbf{g}_{out} \leftarrow \sum_{i=1}^{N} \nabla \ell_i(\boldsymbol{\theta}_{in})$
5:     $\boldsymbol{\theta}_{out} \leftarrow \boldsymbol{\theta}_{in}$
6:     **for** $t = 1, 2, \ldots, m$ **do**
7:         Randomly pick $i \in \{1, 2, \ldots, N\}$
8:         $\boldsymbol{\theta}_{in} = \mathbf{m}_{in} + \boldsymbol{\epsilon}$ with $\boldsymbol{\epsilon} \sim \mathcal{N}(0, \mathbf{I})$
9:         $\mathbf{g}_{in} \leftarrow \nabla \ell_i(\boldsymbol{\theta}_{in}) - \nabla \ell_i(\boldsymbol{\theta}_{out}) + \frac{1}{N}\mathbf{g}_{out}$
10:        $\mathbf{m}_{in} \leftarrow \mathbf{m}_{in} - \eta \mathbf{g}_{in}$
11:     **end for**
12: **end while**

obtain the following expression for the update of $\boldsymbol{\lambda}_{in}$,

$$\boldsymbol{\lambda}_{in} \leftarrow (1 - \eta)\boldsymbol{\lambda}_{in} - \eta N \big[ \widetilde{\nabla}\mathcal{L}_i(\boldsymbol{\lambda}_{in}) - \widetilde{\nabla}\mathcal{L}_i(\boldsymbol{\lambda}_{out}) \\ + \frac{1}{N}\sum_{j=1}^{N} \widetilde{\nabla}\mathcal{L}_j(\boldsymbol{\lambda}_{out}) \big]. \quad (17)$$

There is a one-to-one correspondence between this and Eq. 3. Essentially, all instances of $\boldsymbol{\theta}$ are replaced by $\boldsymbol{\lambda}$ and gradients are replaced by natural-gradients; a minor difference is in the use of learning rate. Due to these similarities, we can write a SVRG-style algorithm shown in Alg. 2.

Our main claim in this paper is to propose Alg. 2 as a Bayesian generalization of SVRG and use it to derive new extensions that go beyond SVRG and its existing variants. We start with our first result to show that SVRG can be derived as a special case of Eq. 16 by restricting the form of $q$ to an isotropic Gaussian density $\mathcal{N}(\boldsymbol{\theta}|\mathbf{m}, \mathbf{I})$ with mean $\mathbf{m}$ and covariance set to the identity, shown in Eq. 7.

**Theorem 1** *When we use isotropic-Gaussian densities $q_{in} = \mathcal{N}(\boldsymbol{\theta}|\mathbf{m}_{in}, \mathbf{I})$ and $q_{out} = \mathcal{N}(\boldsymbol{\theta}|\mathbf{m}_{out}, \mathbf{I})$, then Eq. 16 reduces to the following SVRG-like update of $\mathbf{m}_{in}$,*

$$\mathbf{m}_{in} \leftarrow \mathbf{m}_{in} - \eta N \Big[ \mathbb{E}_{q_{in}}[\nabla \ell_i] - \mathbb{E}_{q_{out}}[\nabla \ell_i] + \frac{1}{N}\sum_{j=1}^{N} \mathbb{E}_{q_{out}}[\nabla \ell_j] \Big] \quad (18)$$

A proof is shown in App. B and uses the definitions of $\boldsymbol{\lambda}, \boldsymbol{\mu}$, and $h(\boldsymbol{\theta})$, as well as Bonnet's theorem (Bonnet, 1964) which states that natural gradients can conveniently be computed using the expectation of $\ell_i(\boldsymbol{\theta})$ at sample $\boldsymbol{\theta} \sim q$, that is, $\nabla_{\mathbf{m}}\mathcal{L}_i = \mathbb{E}_q[\nabla \ell_i]$. An algorithm to implement the update is given in Alg. 3 where changes on top of SGD are highlighted in red. The main change is to add Gaussian noise in line 3 and 8 which arises due to the sampling from $q$. Such weight-perturbation is common for variational learning, for instance, variational variants of gradient

descent (Khan & Rue, 2023, Sec. 1.3.1) and Adam (Khan et al., 2018). Due to this connection, we call this algorithm 'Variational SGD with PoCo' or VSGD-PoCo. Except the sampling part, the algorithm is identical to SVRG of Alg. 1.

The following theorem shows that we can recover SVRG by removing the sampling part from VSGD-PoCo.

**Theorem 2** *SVRG (Alg. 1) is equivalent to VSGD-PoCo (Alg. 3) when we set $\boldsymbol{\epsilon} \leftarrow 0$.*

Setting $\boldsymbol{\epsilon} \leftarrow 0$ is equivalent to the delta method used earlier in Eq. 5 to approximate $\mathbb{E}_q[\ell_i] \approx \ell_i(\mathbf{m})$, as shown below:

$$\mathbb{E}_{\mathcal{N}(\boldsymbol{\theta}|\mathbf{m}, \mathbf{I})}[\nabla \ell_i] = \mathbb{E}_{\mathcal{N}(\boldsymbol{\epsilon}|0, \mathbf{I})}[\nabla \ell_i(\mathbf{m} + \boldsymbol{\epsilon})] \approx \nabla \ell_i(\mathbf{m}).$$

By connecting SVRG to PoCo, Thm. 2 offers a new perspective of SVRG as a knowledge-transfer mechanism. Full-batch gradient computation can be interpreted as aggregation of old knowledge, while gradient corrections can be seen a way to use this knowledge to stabilize minibatch-gradient steps. The result provides a direct connection between knowledge transfer and variance reduction.

### 3.3. Going Beyond SVRG

We will now demonstrate that, by using other posterior forms in Eq. 16, we can derive new extensions of SVRG. Any exponential-family form can be used, for instance, we can use the Bernoulli distribution used to train binary neural networks (Meng et al., 2020). A straightforward calculation shows that this yields SVRG-style updates for the Straight-Through Estimator (Bengio et al., 2013). However, for simplicity's sake, we will not discuss such extensions and focus on the simpler Gaussian case which is sufficient to show the application of our generalization.

#### 3.3.1. A NEWTON-LIKE SVRG VARIANT

To derive a Newton-like variant, we use multivariate Gaussians $q = \mathcal{N}(\boldsymbol{\theta}|\mathbf{m}, \mathbf{S}^{-1})$, where $\mathbf{S}$ is the precision matrix (inverse of the covariance). When using such posteriors in Alg. 2, the precision matrix also has to be corrected in addition to the mean which in turn means that Hessians need to be corrected. This is our second main result.

**Theorem 3** *For Gaussian $q_{in} = \mathcal{N}(\mathbf{m}_{in}, \mathbf{S}_{in}^{-1})$, Eq. 17 reduces to a Newton-like update,*

$$\mathbf{m}_{in} = \mathbf{m}_{in} - \eta N \mathbf{S}_{in}^{-1} \Big[ \mathbb{E}_{q_{in}}[\nabla \ell_i] - \mathbb{E}_{q_{out}}[\nabla \ell_i] \\ + \frac{1}{N}\sum_{j=1}^{N} \mathbb{E}_{q_{out}}[\nabla \ell_j] + \mathbf{H}_{out \setminus i}(\mathbf{m}_{in} - \mathbf{m}_{out}) \Big] \quad (19)$$

*where we use a Stochastic Variance-Reduced Hessian (SVRH) estimate as the pre-conditioner*

$$\mathbf{S}_{in} \leftarrow (1 - \eta)\mathbf{S}_{in} + \eta N \big[ \mathbb{E}_{q_{in}}[\nabla^2 \ell_i] + \bar{\mathbf{H}}_{out \setminus i} \big]. \quad (20)$$

---

**Algorithm 4** IVON-PoCoMo: IVON with Posterior Correction and Momentum. IVON-PoCo is obtained by removing momentum. Differences to IVON highlighted in red.

---

**Require:** Learning rates $\{\eta_t\}$, $\beta_1 \in [0,1)$, $\beta_2 \in [0,1)$, $\delta > 0$, $\kappa > 0$, $h_0 > 0$, clip radius $\xi > 0$, mini-batch size $B$, mega-batch size $M$, outer loop learning rate $\rho_1, \rho_2$, refresh rate $\alpha$.
1: Initialize: $\mathbf{m}_{\text{in}} \leftarrow$ (NN weight init), $\mathbf{h} \leftarrow h_0$, $\boldsymbol{\sigma}_{\text{in}} \leftarrow 1/\sqrt{\kappa(\mathbf{h}_{\text{in}} + \delta)}$, $\mathbf{g} \leftarrow 0$, $\mathbf{g}_{\text{out}} \leftarrow 0$, $\mathbf{h}_{\text{out}} \leftarrow 0$
2: **while** not converged **do**
3: $\quad \widehat{\mathbf{g}}_{\text{out}} \leftarrow \frac{1}{M} \sum_{i \in \mathcal{M}} \nabla \ell_i(\boldsymbol{\theta}_{\text{in}}) \quad$ where we sample a mega-batch $\mathcal{M}$ and $\boldsymbol{\theta}_{\text{in}} \sim \mathcal{N}(\mathbf{m}_{\text{in}}, \boldsymbol{\sigma}_{\text{in}}^2)$
4: $\quad \mathbf{g}_{\text{out}} \leftarrow \rho_1 \mathbf{g}_{\text{out}} + (1 - \rho_1)\widehat{\mathbf{g}}_{\text{out}}, \quad$ and $\quad \mathbf{h}_{\text{out}} \leftarrow \rho_2 \mathbf{h}_{\text{out}} + (1 - \rho_2)\widehat{\mathbf{g}}_{\text{out}}(\boldsymbol{\theta}_{\text{in}} - \mathbf{m}_{\text{in}})/\boldsymbol{\sigma}_{\text{in}}^2$
5: $\quad \mathbf{m}_{\text{out}} \leftarrow \mathbf{m}_{\text{in}}, \ \boldsymbol{\sigma}_{\text{out}} \leftarrow \boldsymbol{\sigma}_{\text{in}}$
6: $\quad$ **for** $t = 1, 2, \dots, m$ **do**
7: $\quad\quad$ Sample a mini-batch $\mathcal{B}$, $\boldsymbol{\theta}_{\text{in}} \sim \mathcal{N}(\mathbf{m}_{\text{in}}, \boldsymbol{\sigma}_{\text{in}}^2)$, $\boldsymbol{\theta}_{\text{out}} \sim \mathcal{N}(\mathbf{m}_{\text{out}}, \boldsymbol{\sigma}_{\text{out}}^2)$
8: $\quad\quad \widehat{\mathbf{g}}_{\text{in}} \leftarrow \frac{1}{B} \sum_{i \in \mathcal{B}} \nabla \ell_i(\boldsymbol{\theta}_{\text{in}}) \quad$ and $\quad \widehat{\mathbf{h}}_{\text{in}} \leftarrow \widehat{\mathbf{g}}_{\text{in}}(\boldsymbol{\theta}_{\text{in}} - \mathbf{m}_{\text{in}})/\boldsymbol{\sigma}_{\text{in}}^2$
9: $\quad\quad \widehat{\mathbf{g}}_{\text{out}} \leftarrow \frac{1}{B} \sum_{i \in \mathcal{B}} \nabla \ell_i(\boldsymbol{\theta}_{\text{out}}) \quad$ and $\quad \widehat{\mathbf{h}}_{\text{out}} \leftarrow \widehat{g}_{\text{out}}(\boldsymbol{\theta}_{\text{out}} - \mathbf{m}_{\text{out}})/\boldsymbol{\sigma}_{\text{out}}^2$
10: $\quad\quad \widehat{\mathbf{g}} \leftarrow \widehat{\mathbf{g}}_{\text{in}} - \alpha(\widehat{\mathbf{g}}_{\text{out}} - \mathbf{g}_{\text{out}}) \quad$ and $\quad \widehat{\mathbf{h}} \leftarrow \widehat{\mathbf{h}}_{\text{in}} - \alpha(\widehat{\mathbf{h}}_{\text{out}} - \mathbf{h}_{\text{out}})$
11: $\quad\quad \mathbf{g} \leftarrow \beta_1 \mathbf{g} + (1 - \beta_1)\widehat{\mathbf{g}}$
12: $\quad\quad \mathbf{h} \leftarrow \beta_2 \mathbf{h} + (1 - \beta_2)\widehat{\mathbf{h}} + \frac{1}{2}(1 - \beta_2)^2(\mathbf{h} - \widehat{\mathbf{h}})^2/(\mathbf{h} + \delta)$
13: $\quad\quad \bar{\mathbf{g}} \leftarrow \mathbf{g}/(1 - \beta_1^t)$
14: $\quad\quad \bar{\mathbf{g}} \leftarrow (\bar{\mathbf{g}} + \delta \mathbf{m}_{\text{in}} + \alpha(\mathbf{h}_{\text{out}} - \widehat{\mathbf{h}}_{\text{out}})(\mathbf{m}_{\text{in}} - \mathbf{m}_{\text{out}}))/(\mathbf{h} + \delta)$
15: $\quad\quad \mathbf{m}_{\text{in}} \leftarrow \mathbf{m}_{\text{in}} - \eta_t \, \text{clip}(\bar{\mathbf{g}}, \xi)$
16: $\quad\quad \boldsymbol{\sigma}_{\text{in}} \leftarrow 1/\sqrt{\kappa(\mathbf{h} + \delta)}$
17: $\quad$ **end for**
18: **end while**
19: **return** $\mathbf{m}, \boldsymbol{\sigma}$

---

*Here, $\bar{\mathbf{H}}_{out \setminus i} = \frac{1}{N} \sum_{j=1}^N \mathbb{E}_{q_{out}}[\nabla^2 \ell_j] - \mathbb{E}_{q_{out}}[\nabla^2 \ell_i]$ is the full-batch expected Hessian without $\ell_i$.*

The derivation uses the definition of natural parameter and gradient in Eq. 17 and is given in App. C. The update is an SVRG-style extension of the Variational-Online-Newton (VON) algorithm of Khan et al. (2018). An algorithm based on this, which we call VON-PoCo, is in Alg. 5.

To our knowledge no other Newton variant of SVRG implements Hessian corrections similarly to SVRH. Instead, most works on Newton steps only use corrections for the gradient and do not do any Hessian correction at all (Derezinski, 2025; Sadiev et al., 2024; Garg et al., 2025; Sun et al., 2025). Some works have used the term $\mathbf{H}_{out \setminus i}(\mathbf{m}_{\text{in}} - \mathbf{m}_{\text{out}})$ in Eq. 19. For instance, Chayti et al. (2024, Eqs. 11–12) derive it via cubic-Newton and Gower et al. (2018) use it to propose the SVRG2 algorithm. The term has an intuitive interpretation as forcing the inner iterate to stay close to the most recent outer iterate, similar to proximal terms in federated learning.

Finally, we stress that such Newton-like extensions do not automatically emerge when SVRG is naively applied to a Bayesian algorithm. The natural-gradients used in PoCo are crucial to derive these extensions.

### 3.3.2. ADAM-LIKE UPDATES WITH IVON-POCO

In practice, full covariances like in VON-PoCo are often infeasible. Therefore, diagonal Gaussians $q = \mathcal{N}(\boldsymbol{\theta} \mid \mathbf{m}, \text{diag}(\mathbf{s})^{-1})$ are often used as a more compute-friendly alternative with storage costs just like AdamW, for example. These lead to a Posterior Correction over IVON (Shen et al., 2024) when used in Alg. 2. We show the full algorithm in Alg. 4 and a full derivation is in App. D. There, we use further practical tricks, for example, weight decay, mini-batching, temperature, debiasing, and momentum. The version with momentum is called IVON-PoCoMo, while the version without it is referred to as IVON-PoCo (obtained by setting $\rho_1 = \rho_2 = 1$).

The IVON-PoCo algorithm avoids full-batch computations which are expensive and also impractical in online learning settings like LLM pretraining. Instead, it uses *mega-batches* (Defazio & Bottou, 2019) which can be tens of times the size of the inner loop. These mega-batches can slowly build an estimate of full-batch gradients and Hessians (line 4) via an estimate of the site function. For example, for isotropic Gaussians we can use the following estimate:

$$\hat{q}_{\text{out}} \leftarrow \exp\left(-\boldsymbol{\theta}^\top \mathbf{g}_{\text{out}}\right), \tag{21}$$

$$\text{where } \mathbf{g}_{\text{out}} \leftarrow \rho_1 \mathbf{g}_{\text{out}} + (1 - \rho_1) \sum_{i \in \mathcal{M}} \mathbb{E}_{q_{\text{out}}}[\nabla \ell_i],$$

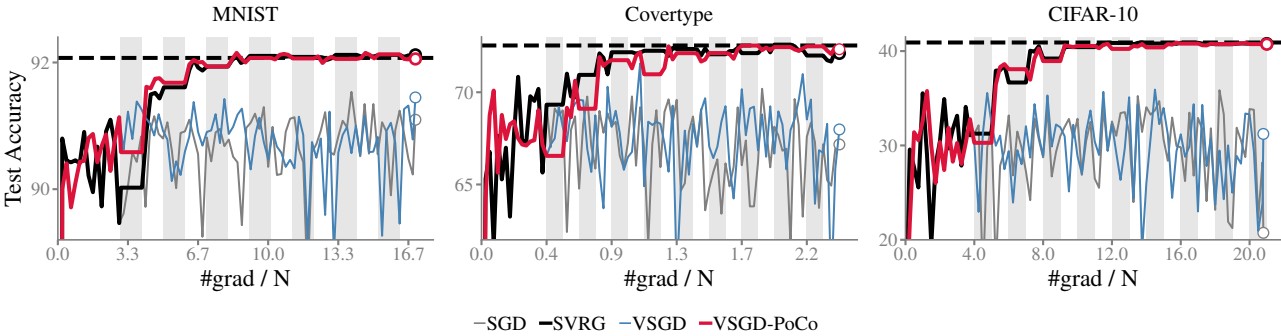

*Figure 3.* The results shown here are in the same settings as Fig. 2 but use first-order methods. We find that SGD and VSGD perform similarly, even though VSGD is a noisy optimization algorithm. Just as SVRG improves SGD, VSGD-PoCo improves over VSGD using gradient corrections that are indicated with the grey bars. Again, performance tends to jump up after a new mega-batch is calculated.

where $\mathcal{M}$ denotes the mega batch. The full-Gaussian case is covered in App. D.1.

The usage of mega-batches deviates from the original SVRG formulation. We can tackle this deviation by down-weighting the contribution of the correction by a certain factor $\alpha < 1$. For instance, we can use the following variant of Eq. 16:

$$q_{\text{in}} \leftarrow q_{\text{in}}^{1-\eta} \hat{q}_{\text{out}}^{\eta\alpha} \exp\left(-\eta N \left[\hat{\ell}_{i|\text{in}} - \alpha\hat{\ell}_{i|\text{out}}\right]\right). \qquad (22)$$

For $\alpha = 0$, this update reduces to standard BLR, while for $\alpha = 1$ we use perfect corrections which are good for full-batch $\hat{q}_{\text{out}}$. When using mega-batches, $\alpha < 1$ could be used and tuned. Interestingly, when applied to isotropic Gaussians, this reduces to $\alpha$-SVRG (Yin et al., 2025), though their motivation to use $\alpha$ does not stem from the use of mega-batches (they appear to use full batches instead). Their motivation is to reduce variance early on via scheduling $\alpha$, starting with a high value. From a Bayesian perspective, the $\hat{q}_{\text{out}}$ estimates are expected to be less useful early on. Thus, it makes more sense to use a 'burn-in' period with $\alpha = 0$ and then turn on $\alpha$. We use a constant $\alpha$ and ablate choices in Fig. 9.

### 3.4. Computational & Memory Requirements

The computation and memory overheads of IVON-PoCoMo are similar to the use of Adam to implement $\alpha$-SVRG. Adam uses an accumulation of squared gradients, and our methods use the reparameterization trick to compute a Hessian estimate (line 4, 8, 9). Both scale with $\mathcal{O}(d)$. Unlike $\alpha$-SVRG, IVON-PoCoMo uses an additional Hessian correction as well, but this does not add a significant cost because the Hessian is already computed. We just need to store an additional $\mathbf{h}_{\text{out}}$ in the outer loop (in line 9), as well as $\boldsymbol{\sigma}_{\text{out}}$ (line 5), which both increase memory by $\Theta(d)$. Similarly to VSGD-PoCo, sampling is added in line 3 and 7. All of these costs are not significant compared to the main overhead which all SVRG-based methods

share. Just like in SVRG and $\alpha$-SVRG, the major overhead is the mega-batch computation and the use of two gradients (line 10). In SVRG, the number of gradient computations is tripled, as the mega-batch gradient is a full-batch gradient that is refreshed after every epoch. In our method, for megabatch size $|\mathcal{M}|$ and number of inner iterations $m$ we thus have an overhead of $2 \cdot n \cdot N + \lfloor \frac{n \cdot N}{m} \rfloor \cdot |\mathcal{M}|$, where $n$ is the number of epochs. When $|\mathcal{M}| = m$ the overhead is the same as SVRG, when $|\mathcal{M}| > m$ it is higher, and when $|\mathcal{M}| < m$ it is lower.

## 4. Experiments & Results

Here, we evaluate our novel methods on a variety of learning problems and architectures. First, we show an exploration of various logistic regression problems in Sec. 4.1, where we find strong improvements when using VSGD-PoCo and IVON-PoCo instead of VSGD and IVON. Then, we perform a thorough exploration of IVON-PoCo and IVON-PoCoMo on various deep learning problems. We discuss language model pretraining on GPT-2 and image classification in Sec. 4.2 and Sec. 4.3 respectively. To the best of our knowledge, there are no such application of SVRG-based methods for GPT-2 (except some that use it as momentum). We provide further results and ablations, for example, on continual pretraining in App. E.

### 4.1. Logistic Regression

SVRG has been highly successful in boosting logistic regression with SGD by accelerating convergence. Our aim here is therefore to evaluate whether VSGD benefits similarly from posterior correction by comparing it to VSGD-PoCo. Afterwards, we move beyond first-order optimizers by comparing IVON and IVON-PoCo which also correct Hessians. Both sets of experiments use the same convex problems which vary in dimensionality and number of data examples.

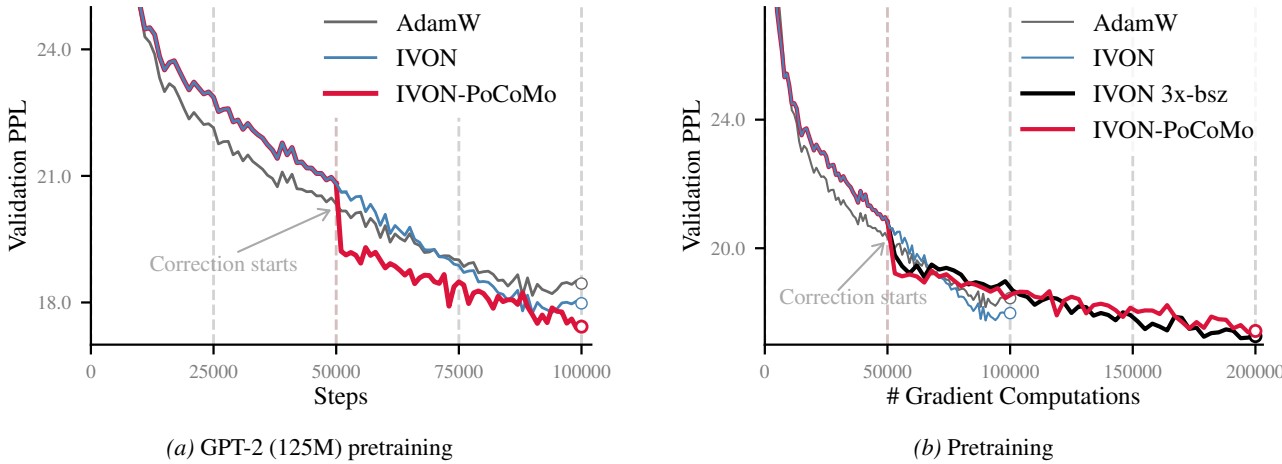

*(a)* GPT-2 (125M) pretraining

*(b)* Pretraining

*Figure 4.* (a) Comparison of our new IVON-PoCoMo (red) with IVON and AdamW when pretraining GPT2-125M from scratch on ca. 50B tokens from OpenWebText. Until 50K steps IVON-PoCo takes the same steps as IVON. Then, correction is started leading to a decrease in loss that also leads to better validation perplexities at the end (17.4, 18.0, 18.4 for IVON-PoCo, IVON, AdamW). (b) When comparing the number of gradient evaluations, IVON-PoCoMo does not converge as fast as IVON, here on the same GPT2-125M pretraining as in Sec. 4.2. Note that the plot only shows validation perplexity calculated every 1,000 steps due to resource limitations. Therefore, the times taken for megabatch calculation, where validation perplexity remains constant, are not shown here.

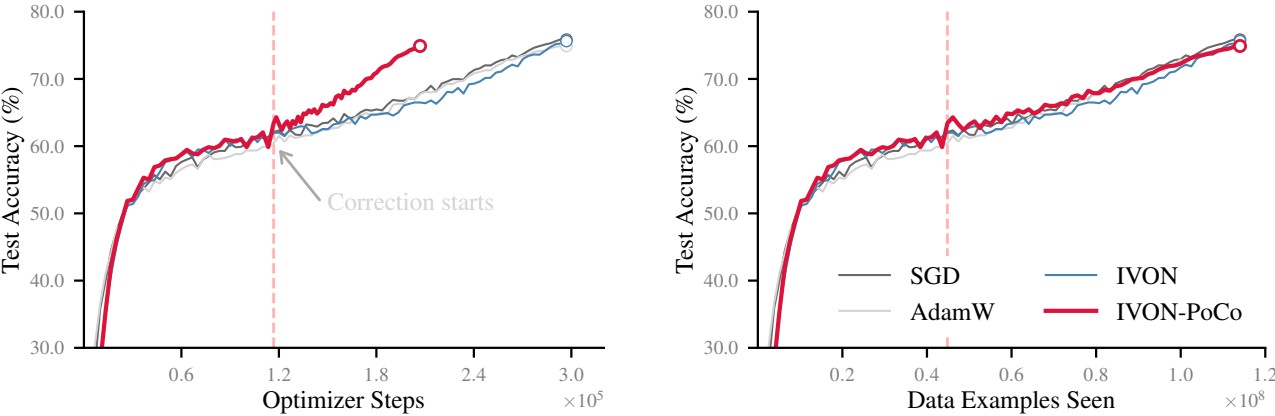

*Figure 5.* Performance on ImageNet for ResNet-50. When comparing by the number of optimization steps (left) IVON-PoCo gives clear improvements but not in terms of data examples seen (right).

We illustrate our results in Fig. 2 and first discuss the performance of first-order methods, where we compare SGD, SVRG, VSGD, and VSGD-PoCo. We use a constant learning rate of 0.01 for all methods with a batch size of 5. For SVRG and VSGD-PoCo we refresh the outer batch every 10,000 steps with a megabatch size of 50,000 after using 20,000 warmup steps. We find that SGD and VSGD provide similar performance, even though VSGD uses random parameter noise. However, both do not converge to the performance obtained at the minimizer, which is indicated by the gray dashed line. SVRG and VSGD-PoCo both reach this performance at similar speed. Consistent with the original work by Johnson & Zhang (2013), SVRG drastically boosts the performance of SGD once the first outer loop is reached, as illustrated by the gray bars, which indicate

when the large outer batch is refreshed. Interestingly, a similarly drastic boost is obtained by using PoCo on top of VSGD.

Next, we discuss results when repeating the same experiment with IVON and IVON-PoCo which is also shown in Fig. 2. Again, we use a constant learning rate and the same mini- and mega-batch sizes as well as 20,000 warmup steps. In addition, we set $\rho_1 = \rho_2 = 1$ (i.e. no momentum) and downweight the extra terms in line 17 of Alg. 4 by 0.01. Again, we find clear boosts when using IVON-PoCo over IVON, starting from the first correction using the first mega-batch. While IVON does not converge to the performance of the minimizer, IVON-PoCo reaches similar accuracy. We also find that IVON and IVON-PoCo converge faster than the first-order methods.

## 4.2. Language Model Pretraining

We apply IVON-PoCoMo to boost IVON training of GPT-2 (125M parameters) from scratch on 50B tokens from the OpenWebText dataset[1]. We follow the set-up from Shen et al. (2024) and train each model for 100,000 steps using AdamW, IVON, and IVON-PoCoMo. For IVON-PoCoMo we use two different configurations: One refreshes a mega-batch with size 10 times the minibatch size after 10 inner steps, starting correction after 50,000 steps, and another where we start correction at varying points, namely, after 30,000, 50,000, and 70,000 steps with a megabatch factor and refresh rate of 20 as opposed to 10. Details on these experiments are found in App. F.2.

Results are shown in Fig. 4a and Fig. 8a, respectively. We find that IVON-PoCoMo provides improvements in terms of final validation perplexity in both settings. Every time correction is added, a strong immediate drop in validation perplexity is observed, even in later and early stages of training. Still, these improvements do not translate into tangible speed-ups of training. IVON-PoCoMo requires more gradient computations (and compute time) to reach similar results as IVON and yields similar results as tripling the batch size, shown in Fig. 4b. The latter has the same compute cost as IVON-PoCoMo and is often used in LLM pretraining (Project Apertus et al., 2025; Singh et al., 2026). Still, we are hopeful that speed boosts can be obtained in the future.

## 4.3. Image Classification with ResNets

Next, we provide further exploration of our methods in deep learning settings. SVRG and SVRG-based methods have often struggled with these, especially with training ResNets on ImageNet (Defazio & Bottou, 2019). In the first experiment, we compare IVON-PoCo to SGD, IVON, and AdamW. We train all methods for the same number of data examples. This means that we stop training earlier for IVON-PoCo, because examples are already seen in mega-batch computations but not reused again for mini-batches.

Results for this experiment are shown in Fig. 5. In particular, we find in the left panel that IVON-PoCo provides clear improvements when just counting optimization steps, as is done in the recent work by Yin et al. (2025). As soon as correction is added, there is a direct increase in test accuracy that persists until the end of training is reached. All other methods perform similarly at the end of training.

However, the right panel also shows that it is hard to beat the baselines when counting the number of data examples seen or the number of gradient evaluations taken, which is consistent with previous findings on SVRG. There, we

find that IVON-PoCo gives a sudden strong initial boost when correction is added but ends up performing similarly as SGD, AdamW, and IVON. A similar result is also found in App. E.1, where we train a ResNet-20 on CIFAR-10 and also compare it to $\alpha$-SVRG. Details are in App. F.1.

## 5. Conclusion

In this paper, we present a surprising, previously unknown connection between SVRG and Bayes. The connection views SVRG as a knowledge adaptation method that is generalized by Posterior Correction. Using Posterior Correction, we derive novel algorithms for various posterior structures, notably diagonal Gaussians. We find that these algorithms provide strong improvements on logistic regression tasks and can also boost deep learning training. However, for deep learning they do not yet provide similar speed-ups as on convex problems. Our work lays a foundation for future work that can build on the connection between SVRG and Bayes to boost deep learning.

## Acknowledgements

This research work has been funded by the German Federal Ministry of Research, Technology and Space and the Hessian Ministry of Higher Education, Research, Science and the Arts within their joint support of the National Research Center for Applied Cybersecurity ATHENE.

MEK and TM are supported by the Bayes duality project, JST CREST Grant Number JPMJCR211.

## Impact Statement

This paper presents work whose goal is to advance the field of machine learning. There are many potential societal consequences of our work, none of which we feel must be specifically highlighted here.

## Author Contribution

Authors: Nico Daheim (ND), Thomas Möllenhoff (TM), Ming Liang Ang (MLA), Mohammad Emtiyaz Khan (MEK).

TM suggested a potential connection between SVRG and PoCo which was then derived by MEK in consultation with both ND and TM. ND did the initial implementation and ablations, which was then improved by both ND and TM together. ND implemented and tuned all LLM experiments, while TM implemented logistic regression and image classification experiments. MLA helped tune and run image classification experiments. The paper was written by all authors.

---

[1]https://huggingface.co/datasets/Skylion007/openwebtext

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

## A. Derivation of Eq. 17

We start by noting that EF distributions can be written as an exponential form $q(\boldsymbol{\lambda}) \propto h(\boldsymbol{\theta}) \exp(\boldsymbol{\lambda}^\top \mathbf{T}(\boldsymbol{\theta}))$. For this derivation, we will assume the base measure $h(\boldsymbol{\theta}) = 1$. An extension is given in the next section. Using the definitions of $q_{\text{in}}$, $\hat{q}_{\text{out}}$, and $q_{\text{out}}$, we can rewrite Eq. 16 as shown below,

$$q_{\text{in}} \leftarrow q_{\text{in}}^{1-\eta} \hat{q}_{\text{out}}^{\eta} \exp\left(-\eta N \left[\hat{\ell}_{i|\text{in}} - \hat{\ell}_{i|\text{out}}\right]\right)$$

$$\implies \exp(\boldsymbol{\lambda}_{\text{in}}^\top \mathbf{T}(\boldsymbol{\theta})) \leftarrow \exp(\boldsymbol{\lambda}_{\text{in}}^\top \mathbf{T}(\boldsymbol{\theta}))^{1-\eta} \exp(\boldsymbol{\lambda}_{\text{out}}^\top \mathbf{T}(\boldsymbol{\theta}))^{\eta} \exp\left(-\eta N \left[\hat{\ell}_{i|\text{in}} - \hat{\ell}_{i|\text{out}}\right]\right) \tag{23}$$

Then, we take the logarithm to get rid of the exponentials and write this as an update over the space of loss functions,

$$\boldsymbol{\lambda}_{\text{in}}^\top \mathbf{T}(\boldsymbol{\theta}) \leftarrow (1-\eta)\boldsymbol{\lambda}_{\text{in}}^\top \mathbf{T}(\boldsymbol{\theta}) + \eta \boldsymbol{\lambda}_{\text{out}}^\top \mathbf{T}(\boldsymbol{\theta}) - \eta N \left[\hat{\ell}_{i|\text{in}} - \hat{\ell}_{i|\text{out}}\right] \tag{24}$$

Next, we use the fact that $\hat{\ell}_{i|\text{in}} = \widetilde{\nabla}\mathcal{L}_i(\boldsymbol{\lambda}_{\text{in}})^\top \mathbf{T}(\boldsymbol{\theta})$ and $\hat{\ell}_{i|\text{out}} = \widetilde{\nabla}\mathcal{L}_i(\boldsymbol{\lambda}_{\text{out}})^\top \mathbf{T}(\boldsymbol{\theta})$ to get

$$\boldsymbol{\lambda}_{\text{in}}^\top \mathbf{T}(\boldsymbol{\theta}) \leftarrow (1-\eta)\boldsymbol{\lambda}_{\text{in}}^\top \mathbf{T}(\boldsymbol{\theta}) + \eta \boldsymbol{\lambda}_{\text{out}}^\top \mathbf{T}(\boldsymbol{\theta}) - \eta N \left[\widetilde{\nabla}\mathcal{L}_i(\boldsymbol{\lambda}_{\text{in}})^\top \mathbf{T}(\boldsymbol{\theta}) - \widetilde{\nabla}\mathcal{L}_i(\boldsymbol{\lambda}_{\text{out}})^\top \mathbf{T}(\boldsymbol{\theta})\right] \tag{25}$$

$$\implies \boldsymbol{\lambda}_{\text{in}}^\top \mathbf{T}(\boldsymbol{\theta}) \leftarrow ((1-\eta)\boldsymbol{\lambda}_{\text{in}}^\top + \eta \boldsymbol{\lambda}_{\text{out}})^\top \mathbf{T}(\boldsymbol{\theta}) - \eta N \left[\widetilde{\nabla}\mathcal{L}_i(\boldsymbol{\lambda}_{\text{in}}) - \widetilde{\nabla}\mathcal{L}_i(\boldsymbol{\lambda}_{\text{out}})\right]^\top \mathbf{T}(\boldsymbol{\theta}). \tag{26}$$

This can be equivalently written as an update over the space of natural parameter $\boldsymbol{\lambda}_{\text{in}}$, as shown below:

$$\boldsymbol{\lambda}_{\text{in}} \leftarrow (1-\eta)\boldsymbol{\lambda}_{\text{in}} + \eta \boldsymbol{\lambda}_{\text{out}} - \eta N \left[\widetilde{\nabla}\mathcal{L}_i(\boldsymbol{\lambda}_{\text{in}}) - \widetilde{\nabla}\mathcal{L}_i(\boldsymbol{\lambda}_{\text{out}})\right]. \tag{27}$$

Finally, from Eq. 13, it follows that the natural parameter $\boldsymbol{\lambda}_{\text{out}} = -\sum_{i=1}^{N} \widetilde{\nabla}\mathcal{L}_i(\boldsymbol{\lambda}_{\text{out}})$. Using this in the update above, we get the desired result shown below:

$$\boldsymbol{\lambda}_{\text{in}} \leftarrow (1-\eta)\boldsymbol{\lambda}_{\text{in}} - \eta N \left[\widetilde{\nabla}\mathcal{L}_i(\boldsymbol{\lambda}_{\text{in}}) - \widetilde{\nabla}\mathcal{L}_i(\boldsymbol{\lambda}_{\text{out}}) + \frac{1}{N}\sum_{j=1}^{N} \widetilde{\nabla}\mathcal{L}_j(\boldsymbol{\lambda}_{\text{out}})\right]. \tag{28}$$

## B. Proof of Thm. 1

The posterior correction in Eq. 16 assumes that the base measure $h(\boldsymbol{\theta})$ is constant. Unfortunately, this is violated for isotropic Gaussian, so we need to write a version of PoCo which allows for non-constant measures. Fortunately, this follows straightforwardly by using the BLR where the base measure is explicitly included. This is given in Khan & Rue (2023, Eq. 6), which can be written as follows in the distribution form,

$$q \leftarrow q^{1-\eta} \prod_{i=0}^{N} \exp\left(-\eta \hat{\ell}_i\right) \exp\left(-\eta \mathbf{T}(\boldsymbol{\theta})^\top \widetilde{\nabla}\mathbb{E}_q[\log h]\right). \tag{29}$$

Khan (2025, App. B) presents a posterior correction version for non-constant base measure. The PoCo update in Eq. 17 is then modified to the following:

$$\boldsymbol{\lambda}_{\text{in}} \leftarrow (1-\eta)\boldsymbol{\lambda}_{\text{in}} - \eta N \left[\widetilde{\nabla}\mathcal{L}_i(\boldsymbol{\lambda}_{\text{in}}) - \widetilde{\nabla}\mathcal{L}_i(\boldsymbol{\lambda}_{\text{out}}) + \frac{1}{N}\sum_{j=1}^{N} \widetilde{\nabla}\mathcal{L}_j(\boldsymbol{\lambda}_{\text{out}}) + \frac{1}{N}\widetilde{\nabla}\mathbb{E}_{q_{\text{in}}}[\log h]\right]. \tag{30}$$

As shown in Eq. 7, for isotropic Gaussian, we have $h(\boldsymbol{\theta}) = \exp(-\boldsymbol{\theta}^\top \boldsymbol{\theta}/2)$. Therefore, $\mathbb{E}_q[\log h] = -\mathbf{m}^\top \mathbf{m}/2 + \text{const.}$. We can now use this to prove the theorem.

**Theorem 1** *For isotropic-Gaussian $q$, Eq. 17 reduces to the following SVRG-like update of the mean $\mathbf{m}_{in}$,*

$$\mathbf{m}_{in} \leftarrow \mathbf{m}_{in} - \eta N \left[\mathbb{E}_{q_{in}}[\nabla \ell_i] - \mathbb{E}_{q_{out}}[\nabla \ell_i] + \frac{1}{N}\sum_{j=1}^{N} \mathbb{E}_{q_{out}}[\nabla \ell_j]\right]. \tag{15}$$

**Proof:** The proof follows directly by using the definition of the isotropic Gaussian $q = \mathcal{N}(\boldsymbol{\theta} \mid \mathbf{m}, \mathbf{I})$, where $\boldsymbol{\lambda} = \boldsymbol{\mu} = \mathbf{m}$. We start with Eq. 17 and then use these quantities, as well as Bonnet's theorem to arrive at:

$$\boldsymbol{\lambda}_{\text{in}} \leftarrow (1-\eta)\boldsymbol{\lambda}_{\text{in}} - \eta N\left[\widetilde{\nabla}\mathcal{L}_i(\boldsymbol{\lambda}_{\text{in}}) - \widetilde{\nabla}\mathcal{L}_i(\boldsymbol{\lambda}_{\text{out}}) + \frac{1}{N}\sum_{j=1}^{N}\widetilde{\nabla}\mathcal{L}_j(\boldsymbol{\lambda}_{\text{out}}) + \frac{1}{N}\widetilde{\nabla}\mathbb{E}_{q_{\text{in}}}[\log h]\right]$$

$$\stackrel{\text{Def.}}{\Longrightarrow} \mathbf{m}_{\text{in}} \leftarrow (1-\eta)\mathbf{m}_{\text{in}} - \eta N\left[\widetilde{\nabla}\mathcal{L}_i(\boldsymbol{\lambda}_{\text{in}}) - \widetilde{\nabla}\mathcal{L}_i(\boldsymbol{\lambda}_{\text{out}}) + \frac{1}{N}\sum_{j=1}^{N}\widetilde{\nabla}\mathcal{L}_j(\boldsymbol{\lambda}_{\text{out}}) + \frac{1}{N}\widetilde{\nabla}\mathbb{E}_{q_{\text{in}}}[\log h]\right]$$

$$\stackrel{\text{Eq. 8}}{\Longrightarrow} \mathbf{m}_{\text{in}} \leftarrow (1-\eta)\mathbf{m}_{\text{in}} - \eta N\left[\nabla_{\boldsymbol{\mu}_{\text{in}}}\mathcal{L}_i(\boldsymbol{\mu}_{\text{in}}) - \nabla_{\boldsymbol{\mu}_{\text{out}}}\mathcal{L}_i(\boldsymbol{\mu}_{\text{out}}) + \frac{1}{N}\sum_{j=1}^{N}\nabla_{\boldsymbol{\mu}_{\text{out}}}\mathcal{L}_j(\boldsymbol{\mu}_{\text{out}}) + \frac{1}{N}\widetilde{\nabla}_{\boldsymbol{\mu}_{\text{in}}}\mathbb{E}_{q_{\text{in}}}[\log h]\right] \quad (31)$$

$$\stackrel{\text{Bonnet}}{\Longrightarrow} \mathbf{m}_{\text{in}} \leftarrow (1-\eta)\mathbf{m}_{\text{in}} - \eta N\left[\mathbb{E}_{q_{\text{in}}}[\nabla_{\boldsymbol{\theta}}\ell_i(\boldsymbol{\theta})] - \mathbb{E}_{q_{\text{out}}}[\nabla_{\boldsymbol{\theta}}\ell_i(\boldsymbol{\theta})] + \frac{1}{N}\sum_{j=1}^{N}[\mathbb{E}_{q_{\text{out}}}\nabla_{\boldsymbol{\theta}}\ell_j(\boldsymbol{\theta}) - \frac{1}{N}\mathbf{m}_{\text{in}}]\right].$$

$$\Longrightarrow \mathbf{m}_{\text{in}} \leftarrow \mathbf{m}_{\text{in}} - \eta N\left[\mathbb{E}_{q_{\text{in}}}[\nabla_{\boldsymbol{\theta}}\ell_i(\boldsymbol{\theta})] - \mathbb{E}_{q_{\text{out}}}[\nabla_{\boldsymbol{\theta}}\ell_i(\boldsymbol{\theta})] + \frac{1}{N}\sum_{j=1}^{N}[\mathbb{E}_{q_{\text{out}}}\nabla_{\boldsymbol{\theta}}\ell_j(\boldsymbol{\theta})]\right].$$

$\square$

## C. Derivation of the Newton-like SVRG Extension

We start by writing $q$ in an exponential-family form which in this case is convenient to write in terms of the precision $\mathbf{S} = \boldsymbol{\Sigma}^{-1}$. This is shown below with sufficient statistics highlighted in red,

$$\mathcal{N}(\boldsymbol{\theta}|\mathbf{m}, \mathbf{S}^{-1}) \propto \exp\left[\mathbf{m}^{\top}\mathbf{S}\boldsymbol{\theta} + \text{Tr}\left((-\tfrac{1}{2}\mathbf{S})\boldsymbol{\theta}\boldsymbol{\theta}^{\top}\right)\right]. \quad (32)$$

This is a log-linear form for a sufficient-statistics vector of $\boldsymbol{\theta}$ and $\boldsymbol{\theta}\boldsymbol{\theta}^{\top}$. We denote the natural parameter $\boldsymbol{\lambda} = (\mathbf{Sm}, -\tfrac{1}{2}\mathbf{S})$ consisting of two elements: a vector and a square matrix. The natural gradient can be written in terms of the gradient and Hessian of $\ell_i$ (Khan & Rue, 2023, Eq. 10-11),

$$\widetilde{\nabla}\mathcal{L}_i(\boldsymbol{\lambda}) = \mathbb{E}_q\left[\left(\nabla\ell_i - \nabla^2\ell_i\mathbf{m}, \ \tfrac{1}{2}\nabla^2\ell_i\right)\right]. \quad (33)$$

This also has two elements, and uses gradient and Hessian computed at samples from $q$.

We will denote the $\boldsymbol{\lambda}_{\text{in}} = (\mathbf{S}_{\text{in}}\mathbf{m}_{\text{in}}, -\tfrac{1}{2}\mathbf{S}_{\text{in}})$ and $\boldsymbol{\lambda}_{\text{out}} = (\mathbf{S}_{\text{out}}\mathbf{m}_{\text{out}}, -\tfrac{1}{2}\mathbf{S}_{\text{out}})$. We first write the update for the second entry of $\boldsymbol{\lambda}_{\text{in}}$ which is $-\tfrac{1}{2}\mathbf{S}_{\text{in}}$,

$$\mathbf{S}_{\text{new}} \leftarrow (1-\eta)\mathbf{S}_{\text{in}} + \eta N\left[\mathbb{E}_{q_{\boldsymbol{\lambda}_{\text{in}}}}[\nabla^2\ell_i] - \mathbb{E}_{q_{\boldsymbol{\lambda}_{\text{out}}}}[\nabla^2\ell_i] + \frac{1}{N}\sum_{j=1}^{N}\mathbb{E}_{q_{\boldsymbol{\lambda}_{\text{out}}}}[\nabla^2\ell_j]\right]. \quad (34)$$

We denoted the new value as $\mathbf{S}_{\text{new}}$ to differentiate it with the old value. This will be useful to simplify the update for the

first entry $\mathbf{S}_{\mathrm{in}}\mathbf{m}_{\mathrm{in}}$ which we show below,

$$
\mathbf{S}_{\mathrm{new}}\mathbf{m}_{\mathrm{new}} = (1-\eta)\mathbf{S}_{\mathrm{in}}\mathbf{m}_{\mathrm{in}} - \eta N \left[ \mathbb{E}_{q_{\boldsymbol{\lambda}_{\mathrm{in}}}}[\nabla\ell_i - \nabla^2\ell_i\mathbf{m}_{\mathrm{in}}] - \mathbb{E}_{q_{\boldsymbol{\lambda}_{\mathrm{out}}}}[\nabla\ell_i - \nabla^2\ell_i\mathbf{m}_{\mathrm{out}}] \right.
$$

$$
\left. + \frac{1}{N}\sum_{j=1}^{N}\mathbb{E}_{q_{\boldsymbol{\lambda}_{\mathrm{out}}}}[\nabla\ell_j - \nabla^2\ell_j\mathbf{m}_{\mathrm{out}}] \right]
$$

$$
= \left\{ \mathbf{S}_{\mathrm{new}} - \eta N \left[ \mathbb{E}_{q_{\boldsymbol{\lambda}_{\mathrm{in}}}}[\nabla^2\ell_i] - \mathbb{E}_{q_{\boldsymbol{\lambda}_{\mathrm{out}}}}[\nabla^2\ell_i] + \frac{1}{N}\sum_{j=1}^{N}\mathbb{E}_{q_{\boldsymbol{\lambda}_{\mathrm{out}}}}[\nabla^2\ell_j] \right] \right\} \mathbf{m}_{\mathrm{in}} +
$$

$$
- \eta N \left[ \mathbb{E}_{q_{\boldsymbol{\lambda}_{\mathrm{in}}}}[\nabla\ell_i - \nabla^2\ell_i\mathbf{m}_{\mathrm{in}}] - \mathbb{E}_{q_{\boldsymbol{\lambda}_{\mathrm{out}}}}[\nabla\ell_i - \nabla^2\ell_i\mathbf{m}_{\mathrm{out}}] \right.
$$

$$
\left. + \frac{1}{N}\sum_{j=1}^{N}\mathbb{E}_{q_{\boldsymbol{\lambda}_{\mathrm{out}}}}[\nabla\ell_j - \nabla^2\ell_j\mathbf{m}_{\mathrm{out}}] \right]
$$

$$
= \mathbf{S}_{\mathrm{new}}\mathbf{m}_{\mathrm{in}} - \eta N \left[ \cancel{\mathbb{E}_{q_{\boldsymbol{\lambda}_{\mathrm{in}}}}[\nabla^2\ell_i]\mathbf{m}_{\mathrm{in}}} - \mathbb{E}_{q_{\boldsymbol{\lambda}_{\mathrm{out}}}}[\nabla^2\ell_i]\mathbf{m}_{\mathrm{in}} + \frac{1}{N}\sum_{j=1}^{N}\mathbb{E}_{q_{\boldsymbol{\lambda}_{\mathrm{out}}}}[\nabla^2\ell_j]\mathbf{m}_{\mathrm{in}} \right.
\tag{35}
$$

$$
\left. + \mathbb{E}_{q_{\boldsymbol{\lambda}_{\mathrm{in}}}}[\nabla\ell_i - \cancel{\nabla^2\ell_i\mathbf{m}_{\mathrm{in}}}] - \mathbb{E}_{q_{\boldsymbol{\lambda}_{\mathrm{out}}}}[\nabla\ell_i - \nabla^2\ell_i\mathbf{m}_{\mathrm{out}}] + \frac{1}{N}\sum_{j=1}^{N}\mathbb{E}_{q_{\boldsymbol{\lambda}_{\mathrm{out}}}}[\nabla\ell_j - \nabla^2\ell_j\mathbf{m}_{\mathrm{out}}] \right]
$$

$$
= \mathbf{S}_{\mathrm{new}}\mathbf{m}_{\mathrm{in}} - \eta N \left[ \mathbb{E}_{q_{\boldsymbol{\lambda}_{\mathrm{in}}}}[\nabla\ell_i] - \mathbb{E}_{q_{\boldsymbol{\lambda}_{\mathrm{out}}}}[\nabla\ell_i] + \frac{1}{N}\sum_{j=1}^{N}\mathbb{E}_{q_{\boldsymbol{\lambda}_{\mathrm{out}}}}[\nabla\ell_j] - \mathbb{E}_{q_{\boldsymbol{\lambda}_{\mathrm{out}}}}[\nabla^2\ell_i]\mathbf{m}_{\mathrm{in}} \right.
$$

$$
\left. + \frac{1}{N}\sum_{j=1}^{N}\mathbb{E}_{q_{\boldsymbol{\lambda}_{\mathrm{out}}}}[\nabla^2\ell_j]\mathbf{m}_{\mathrm{in}} + \mathbb{E}_{q_{\boldsymbol{\lambda}_{\mathrm{out}}}}[\nabla^2\ell_i\mathbf{m}_{\mathrm{out}}] - \frac{1}{N}\sum_{j=1}^{N}\mathbb{E}_{q_{\boldsymbol{\lambda}_{\mathrm{out}}}}[\nabla^2\ell_j\mathbf{m}_{\mathrm{out}}] \right]
$$

$$
= \mathbf{S}_{\mathrm{new}}\mathbf{m}_{\mathrm{in}} - \eta N \left[ \mathbb{E}_{q_{\boldsymbol{\lambda}_{\mathrm{in}}}}[\nabla\ell_i] - \mathbb{E}_{q_{\boldsymbol{\lambda}_{\mathrm{out}}}}[\nabla\ell_i] + \frac{1}{N}\sum_{j=1}^{N}\mathbb{E}_{q_{\boldsymbol{\lambda}_{\mathrm{out}}}}[\nabla\ell_j] \right.
$$

$$
\left. \left( \frac{1}{N}\sum_{j=1}^{N}\mathbb{E}_{q_{\boldsymbol{\lambda}_{\mathrm{out}}}}[\nabla^2\ell_j] - \mathbb{E}_{q_{\boldsymbol{\lambda}_{\mathrm{out}}}}[\nabla^2\ell_i] \right) (\mathbf{m}_{\mathrm{in}} - \mathbf{m}_{\mathrm{out}}) \right]
$$

An algorithm can be conveniently written by defining the following outer-loop quantities using the output $\boldsymbol{\lambda}_{\mathrm{in}}$ of the inner loop,

$$
\mathbf{g}_{\mathrm{out}} \leftarrow \sum_{j=1}^{N}\mathbb{E}_{q_{\mathrm{in}}}[\nabla\ell_j], \qquad \mathbf{H}_{\mathrm{out}} \leftarrow \sum_{j=1}^{N}\mathbb{E}_{q_{\mathrm{in}}}[\nabla^2\ell_j].
\tag{36}
$$

We then set $\mathbf{m}_{\mathrm{out}} \leftarrow \mathbf{m}_{\mathrm{in}}$ and $\mathbf{S}_{\mathrm{out}} \leftarrow \mathbf{S}_{\mathrm{in}}$, and the corresponding natural parameter to be $\boldsymbol{\lambda}_{\mathrm{out}}$. Using these, we can write the updates in the inner loop as follows (strictly in this order),

$$
\begin{aligned}
\mathbf{g}_{\mathrm{in}} &\leftarrow \mathbb{E}_{q_{\mathrm{in}}}[\nabla\ell_i] - \mathbb{E}_{q_{\mathrm{out}}}[\nabla\ell_i] + \mathbf{g}_{\mathrm{out}}/N \\
\mathbf{H}_{\mathrm{out}\backslash i} &\leftarrow \mathbf{H}_{\mathrm{out}}/N - \mathbb{E}_{q_{\mathrm{out}}}[\nabla^2\ell_i] \\
\mathbf{H}_{\mathrm{in}} &\leftarrow \mathbb{E}_{q_{\mathrm{in}}}[\nabla^2\ell_i] + \mathbf{H}_{\mathrm{out}\backslash i} \\
\mathbf{S}_{\mathrm{in}} &\leftarrow (1-\eta)\mathbf{S}_{\mathrm{in}} + \eta N \mathbf{H}_{\mathrm{in}} \\
\mathbf{m}_{\mathrm{in}} &\leftarrow \mathbf{m}_{\mathrm{in}} - \eta N \mathbf{S}_{\mathrm{in}}^{-1} \left[ \mathbf{g}_{\mathrm{in}} + \mathbf{H}_{\mathrm{out}\backslash i}(\mathbf{m}_{\mathrm{in}} - \mathbf{m}_{\mathrm{out}}) \right]
\end{aligned}
\tag{37}
$$

These steps are implemented in Alg. 5 by using one Monte-Carlo sample to evaluate the expectations. We highlight in red the new parts added on top of SVRG. We note two useful points regarding the implementation: first, $\mathbf{S}_{\mathrm{in}}$ need to be always updated before $\mathbf{m}_{\mathrm{in}}$, and second, variance is further reduced if different example use different seeds (which is not explicitly written in the algorithm).

**Algorithm 5** VON-PoCo: Variational Online Newton with Posterior Correction

**Initialize:** Number of inner steps $m$, learning rates $\alpha$ and $\beta$

1: Initialize $\mathbf{m}_{\text{in}}, \mathbf{S}_{\text{in}}$
2: **while** not converged **do**
3:      $\mathbf{g}_{\text{out}} \leftarrow \sum_{i=1}^{N} \nabla \ell_i(\boldsymbol{\theta}_{\text{in}})$ where $\boldsymbol{\theta}_{\text{in}} = \mathbf{m}_{\text{in}} + \mathbf{S}_{\text{in}}^{-\frac{1}{2}} \boldsymbol{\epsilon}$ with $\boldsymbol{\epsilon} \sim \mathcal{N}(0, \mathbf{I})$
4:      $\mathbf{H}_{\text{out}} \leftarrow \sum_{i=1}^{N} \nabla^2 \ell_i(\boldsymbol{\theta}_{\text{in}})$
5:      $\boldsymbol{\theta}_{\text{out}} \leftarrow \boldsymbol{\theta}_{\text{in}}, \mathbf{m}_{\text{out}} \leftarrow \mathbf{m}_{\text{in}}$
6:      **for** $t = 1, 2, \ldots, m$ **do**
7:         Randomly pick $i \in \{1, 2, \ldots, N\}$
8:         $\mathbf{g}_{\text{in}} \leftarrow \nabla \ell_i(\boldsymbol{\theta}_{\text{in}}) - \nabla \ell_i(\boldsymbol{\theta}_{\text{out}}) + \frac{1}{N} \mathbf{g}_{\text{out}}$ where $\boldsymbol{\theta}_{\text{in}} = \mathbf{m}_{\text{in}} + \mathbf{S}_{\text{in}}^{-\frac{1}{2}} \boldsymbol{\epsilon}$ with $\boldsymbol{\epsilon} \sim \mathcal{N}(0, \mathbf{I})$
9:         $\mathbf{H}_{\text{out} \backslash i} \leftarrow \frac{1}{N} \mathbf{H}_{\text{out}} - \nabla^2 \ell_i(\boldsymbol{\theta}_{\text{out}})$
10:        $\mathbf{H}_{\text{in}} \leftarrow \nabla^2 \ell_i(\boldsymbol{\theta}_{\text{in}}) + \mathbf{H}_{\text{out} \backslash i}$
11:        $\mathbf{S}_{\text{in}} \leftarrow (1 - \beta) \mathbf{S}_{\text{in}} + \beta N \mathbf{H}_{\text{in}}$
12:        $\mathbf{m}_{\text{in}} \leftarrow \mathbf{m}_{\text{in}} - \alpha N \mathbf{S}_{\text{in}}^{-1} \left[ \mathbf{g}_{\text{in}} + \mathbf{H}_{\text{out} \backslash i} (\mathbf{m}_{\text{in}} - \mathbf{m}_{\text{out}}) \right]$
13:      **end for**
14: **end while**

## D. Derivation of the Adam-like SVRG Extension

To derive the SVRG extension of IVON, we will first write the VB objective in the form used by IVON; see Shen et al. (2024, Eq. 1). Essentially, they use mini-batches $\mathcal{B}$ of size B, and to accommodate this they scale the expected loss by a constant $\kappa \mathbb{E}_q[\ell_i]$. Setting $\kappa = N$ gives back the ERM loss but it can also be set to other values. In IVON, we also treat the regularizer $\ell_0$ explicitly by setting it to the weight decay. It is not merged in the losse $\ell_i$ as in the previous sections. With these changes, Sec. 3.2 can be written as where an explicit natural gradient of $\mathcal{L}_0$ is added,

$$\boldsymbol{\lambda}_{\text{in}} \leftarrow (1 - \eta) \boldsymbol{\lambda}_{\text{in}} - \eta \kappa \left[ \frac{1}{B} \sum_{i \in \mathcal{B}} \left( \widetilde{\nabla} \mathcal{L}_i(\boldsymbol{\lambda}_{\text{in}}) - \widetilde{\nabla} \mathcal{L}_i(\boldsymbol{\lambda}_{\text{out}}) \right) + \frac{1}{\kappa} \sum_{j=1}^{N} \widetilde{\nabla} \mathcal{L}_j(\boldsymbol{\lambda}_{\text{out}}) + \frac{1}{\kappa} \widetilde{\nabla} \mathcal{L}_0(\boldsymbol{\lambda}_{\text{in}}) \right].$$

We then plug in the natural parameter and natural gradients of $q = \mathcal{N}(\boldsymbol{\theta}|\mathbf{m}, \text{diag}(\mathbf{s})^{-1})$, which yields a Newton-like update very similar to Thm. 3.

To derive the IVON-PoCo update, we assume a quadratic regularizer $\ell_0 = s_0 \frac{1}{2} \boldsymbol{\theta}^\top \boldsymbol{\theta}$ with $s_0 > 0$. The VONcorr update can be written as follows where the new parts compared to Eq. 19 are in red,

$$\mathbf{s}_{\text{in}} \leftarrow (1 - \eta) \mathbf{s}_{\text{in}} + \eta \kappa \left[ \frac{1}{B} \sum_{i \in \mathcal{B}} \left( \mathbb{E}_{q_{\boldsymbol{\lambda}_{\text{in}}}}[\nabla^2 \ell_i] - \mathbb{E}_{q_{\boldsymbol{\lambda}_{\text{out}}}}[\nabla^2 \ell_i] \right) + \frac{1}{\kappa} \sum_{j=1}^{N} \mathbb{E}_{q_{\boldsymbol{\lambda}_{\text{out}}}}[\nabla^2 \ell_j] + \frac{s_0}{\kappa} \right],$$

$$\begin{aligned}
\mathbf{m}_{\text{in}} \leftarrow \mathbf{m}_{\text{in}} - \eta \kappa \frac{1}{\mathbf{s}_{\text{in}}} \Bigg[ & \frac{1}{B} \sum_{i \in \mathcal{B}} \left( \mathbb{E}_{q_{\boldsymbol{\lambda}_{\text{in}}}}[\nabla \ell_i] - \mathbb{E}_{q_{\boldsymbol{\lambda}_{\text{out}}}}[\nabla \ell_i] \right) + \frac{1}{\kappa} \sum_{j=1}^{N} \mathbb{E}_{q_{\boldsymbol{\lambda}_{\text{out}}}}[\nabla \ell_j] + \frac{s_0}{\kappa} \mathbf{m}_{\text{in}} \\
& + \left( \frac{1}{\kappa} \sum_{j=1}^{N} \mathbb{E}_{q_{\boldsymbol{\lambda}_{\text{out}}}}[\nabla^2 \ell_j] - \frac{1}{B} \sum_{i \in \mathcal{B}} \mathbb{E}_{q_{\boldsymbol{\lambda}_{\text{out}}}}[\nabla^2 \ell_i] \right) (\mathbf{m}_{\text{in}} - \mathbf{m}_{\text{out}}) \Bigg].
\end{aligned} \quad (38)$$

To write the update in IVON form, we make a few modifications.

1. For weight decay, we tune $\delta = s_0 / \kappa$ directly.

2. We remove $\delta$ from the $\mathbf{s}_{\text{in}}$ update and divide the whole update by $\kappa$. The resulting update is written in terms of $\mathbf{h}_{\text{in}}$ such that $\mathbf{s}_{\text{in}} = \kappa(\mathbf{h}_{\text{in}} + \delta)$ and $\sigma_{\text{in}}^2 = 1/(\kappa(\mathbf{h}_{\text{in}} + \delta))$.

3. We use different learning rate for $\mathbf{m}_{\text{in}}$ and $\mathbf{h}_{\text{in}}$ updates. For $\mathbf{m}_{\text{in}}$, we use a scheduled $\eta_t$ for iteration $t$. For $\mathbf{h}_{\text{in}}$, we use $\beta_2 \in [0, 1)$.

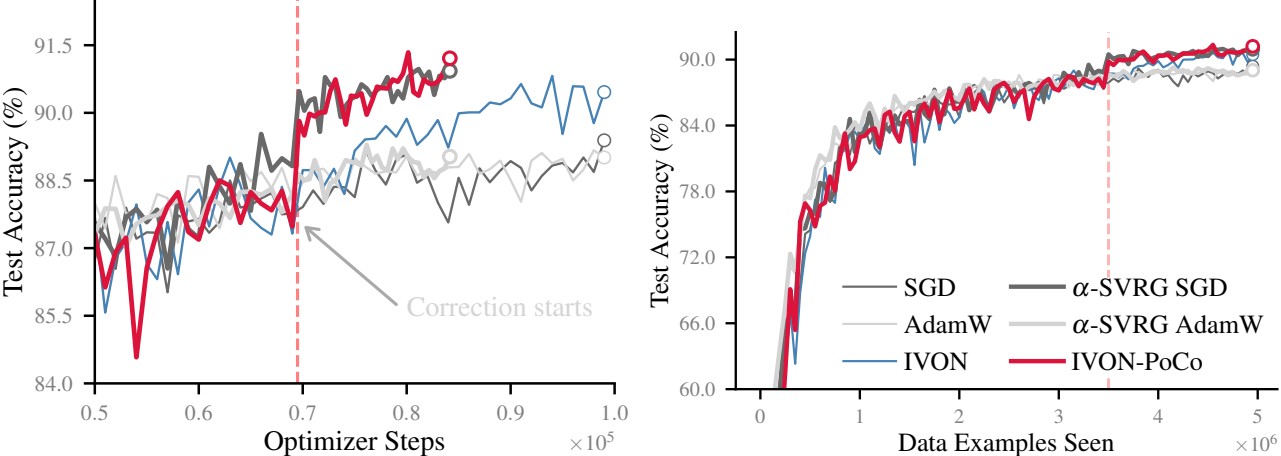

*Figure 6.* Comparison to $\alpha$-SVRG on CIFAR-10 for ResNet-20. IVON-PoCo and $\alpha$-SVRG with SGD give improvements, also when counting the number of data examples seen (right). At the end of training, IVON-PoCo performs better than other methods.

4. We use momentum with learning rate $\beta_1 \in [0, 1)$ and debiasing for $\mathbf{g}$ (but not for $\mathbf{h}$).

5. We add a term for the update of $\mathbf{h}$ which ensures positivity of $\mathbf{h}$. We initialize $\mathbf{h}$ by a scalar constant $h_0 > 0$.

6. The scaling factor $\kappa$ is often set to $N$ but it can be different for cases when the effective number of examples is not immediately clear (for example, for LLM training).

7. We add an additional factor $\alpha$ in front of the outer gradients and Hessian, similarly to $\alpha$-SVRG (Yin et al., 2025).

### D.1. Handling Mega-Batches for Full-Gaussian

Similarly to SVRG, our derivation assumes a full-batch computation in the outer loop, while in practice it is cheaper to use a smaller subset $\mathcal{M}$. However, we could still use a moving average in the outer loop to keep a running estimate of the full batch gradient and Hessian. For full-Gaussians, one way to implement this is by using the following outer-loop distribution where gradient and Hessian averages are maintained,

$$
\begin{aligned}
\hat{q}_{\text{out}} &\leftarrow \exp\left(-\boldsymbol{\theta}^\top \mathbf{g}_{\text{out}} - \tfrac{1}{2}(\boldsymbol{\theta} - \mathbf{m}_{\text{out}})^\top \mathbf{H}_{\text{out}}(\boldsymbol{\theta} - \mathbf{m}_{\text{out}})\right), \\
\text{where} \quad \mathbf{g}_{\text{out}} &\leftarrow \rho_1 \mathbf{g}_{\text{out}} + (1 - \rho_1)\sum_{i \in \mathcal{M}} \mathbb{E}_{q_{\text{out}}}[\nabla \ell_i] \\
\mathbf{H}_{\text{out}} &\leftarrow \rho_2 \mathbf{H}_{\text{out}} + (1 - \rho_2)\sum_{i \in \mathcal{M}} \mathbb{E}_{q_{\text{out}}}[\nabla^2 \ell_i].
\end{aligned}
\tag{39}
$$

It is also possible to implement this through a momentum in the $q$-space (Lin et al., 2023) but we do not pursue this for simplicity's sake.

## E. Additional Results

### E.1. ResNet-20 on CIFAR-10

Here, we train smaller ResNet-20 in Fig. 6. Unlike in the ImageNet training run in Sec. 4.3, we do not anneal the learning rate to zero but to a quarter of the starting learning rate. Then, the variance is not completely removed through learning rate annealing and both $\alpha$-SVRG with SGD and IVON-PoCo bring improvements in accuracy over their respective baseline methods, with IVON-PoCo having the advantage of estimating a variational posterior distribution. All details for the two experiments are in App. F.1.

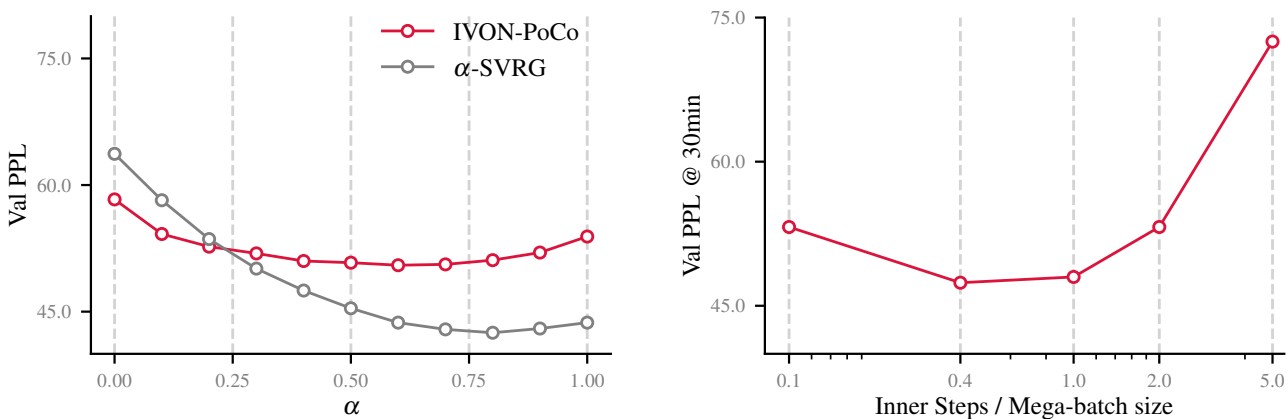

*Figure 7.* (left) Tuning $\alpha$ has a large effect on performance on wikitext103 with a 33M Transformer trained from scratch. (right) In the same setting, increasing the number of refreshes can help but at some point becomes slow, with better performance at the same time budget obtained by fewer refreshes.

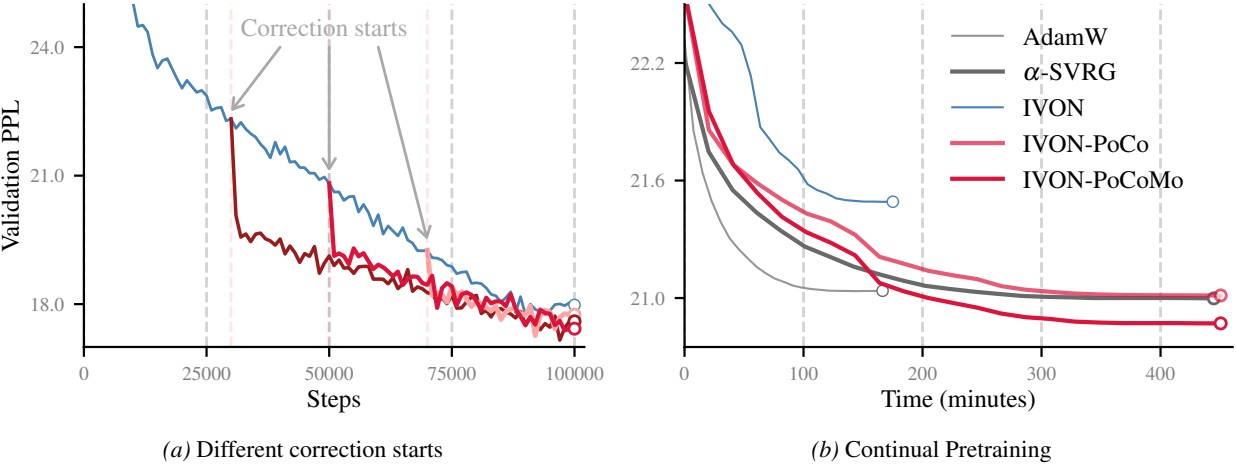

*(a)* Different correction starts        *(b)* Continual Pretraining

*Figure 8.* (Left) (b) Improvements in final results also hold when starting correction earlier or later in training (shown with purple and pink respectively). (Right) IVON-PoCoMo helps stabilize continual pretraining with IVON at higher learning rates but incurs additional computation.

## E.2. Continual Pretraining

We present results on continually pretraining the GPT-125M model from Shen et al. (2024) on 1B tokens from Fineweb-edu (Penedo et al., 2024). We compare AdamW and $\alpha$-SVRG with AdamW against IVON, IVON-PoCo, and IVON-PoCoMo. Both $\alpha$-SVRG and the IVON-PoCo runs use $\alpha = 0.7$, 40 steps for the inner loop, and 1,000 warmup steps without correction. Results are in Fig. 8b. Adding correction both with $\alpha$-SVRG and IVON-PoCo helps but the gap between IVON, which struggles with larger learning rates, and IVON-PoCo is larger. Adding Momentum to IVON-PoCo improves results beyond those of $\alpha$-SVRG and IVON-PoCoMo converges to a better validation perplexity after the same number of steps. When comparing time, $\alpha$-SVRG does not improve over AdamW but IVON-PoCoMo improves over IVON. Details can be found in App. F.3.

## E.3. Finetuning

Here, we use IVON-PoCo for finetuning different Transformers, namely ViT-B/32 (Dosovitskiy et al., 2021), Qwen2.5-0.5B-Instruct (Yang et al., 2025), and Llama-3.1-8B (Grattafiori et al., 2024) on image classification, XSUM, and GSM8k, respectively. For the ViT models we finetune only the image encoder, for Qwen we finetune the entire model, and for Llama-3.1 we use LoRA finetuning (Hu et al., 2022). Results are shown in Table 1 and show that IVON-PoCo can slightly improve final performance over IVON when the models are trained with the same number of optimization steps. Note

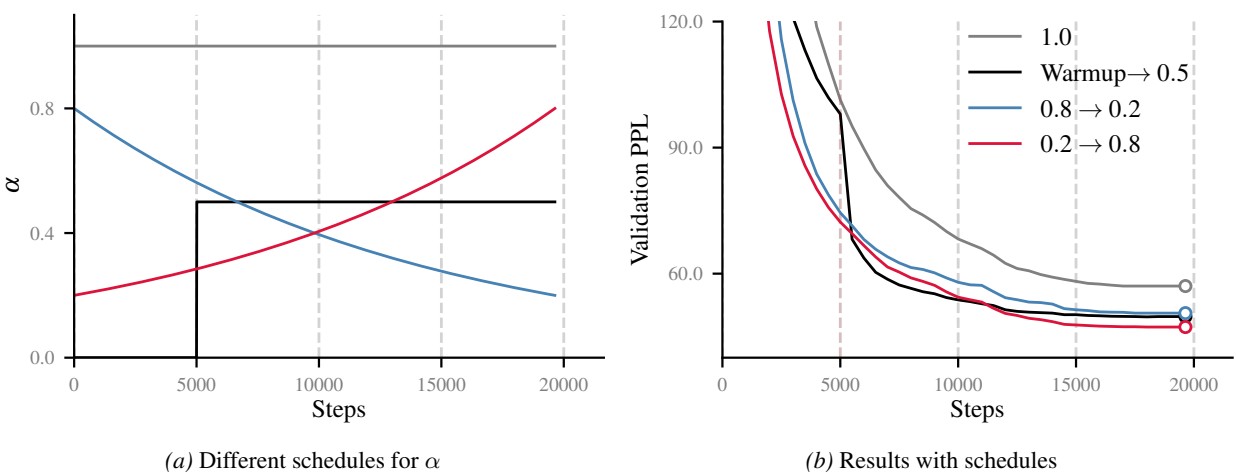

*(a)* Different schedules for $\alpha$                        *(b)* Results with schedules

*Figure 9.* We test different schedules for $\alpha$ that are shown in (a). We find in (b) that increasing $\alpha$ performs better than decreasing it but starting correction only later works well, too. All strategies are better than keeping $\alpha = 1.0$.

*Table 1.* IVON-PoCo provides modest improvements when finetuning ViT-B/32, Qwen2.5-0.5-it and Llama3.1-8B. All methods use the same number of steps. R.-45 stands for RESISC45.

|  | | | ViT | | Qwen | | Llama |
|---|---|---|---|---|---|---|---|
|  | Cars | DTD | GTSRB | R.-45 | XSUM | | GSM8k |
|  | | | Acc. | | R-1 | R-L | Acc. |
| IVON | 79.5 | 72.9 | **99.9** | 95.2 | 48.7 | 23.6 | 30.4 |
| + PoCo | **80.0** | **73.4** | 99.9 | **96.1** | **49.6** | **23.8** | **30.6** |

that runtime is larger for IVON-PoCo due to the additional gradient calculations. Details and hyperparameters for these experiments can be found in App. F.4.

### E.4. Ablations

**Influence of $\alpha$:** We study the influence of $\alpha$ for $\alpha$-SVRG and IVON-PoCo on wikitext103 (Merity et al., 2017) by training a GPT-2-based model with 33M parameters from scratch for one epoch with batch size of 64 and 5000 warmup steps without correction. Results for different values of $\alpha$ are shown in Fig. 7 (left): for both methods using correction improves performance over various values of $\alpha$. Both times the curve has a u-shape indicating that biasing too much towards large batches can harm performance. We provide details for these and the following analysis in App. F.2.

We also evaluate different schedules in the same setting. We compare 5000 warmup steps followed by $\alpha = 0.5$ to keeping $\alpha = 1.0$ for the entire duration as well as increasing from 0.2 to 0.8, for which we advocate, and decreasing from 0.8 to 0.2 which is advocated for in $\alpha$-SVRG. We find in Fig. 9 that increasing $\alpha$ performs best, followed by the warmup-based schedule. Decreasing $\alpha$, nevertheless, performs better than keeping $\alpha = 1.0$.

**Influence of Inner Loop Size** Here, we fix the megabatch size to $50 \cdot 64$ in the same setting as above and vary the number of refreshes. This means that doing more refreshes will turn out to be more expensive for the same number of minibatches seen in inner steps. Fig. 7 (center) shows that this tends to help performance while refreshing too seldomly, shown towards the right of the x-axis, hurts performance due to drift. Nevertheless, refreshing too often also appears to be harmful, indicating that a balance between exploration (via mini-batch gradients) and exploitation (via mega-batch gradients) needs to be struck.

## F. Hyperparameters and Additional Details

### F.1. Image Classification with ResNets

The ImageNet experiments run for 90 epochs, using batch size 384 on 8 GPUs which took around 12-15 hours. The learning rate is annealed to zero using a cosine schedule. For the CIFAR-10 experiments, we train for 100 epochs and

batch size 50 on a single GPU, and each run took around 1-2 hours. As described in the main text, the learning rate is annealed to a quarter of the starting learning rate for all methods.

The hyperparameters for SGD, AdamW and IVON are set identical to the ones used in (Shen et al., 2024), except for CIFAR-10 where we use an ess of $5 \cdot 10^5$. All SVRG methods and IVON-PoCo inherit the hyperparameters from their base algorithm. The variance reduced methods use a megabatch size 50 times larger than the minibatch size on ImageNet and 10 times larger than the minibatch size on CIFAR-10. We tuned $\alpha$ for $\alpha$-SVRG and IVON-PoCo separately. The optimal $\alpha$ is 0.2 for AdamW, 0.4 for SGD and 0.1 for IVON-PoCo. On ImageNet, we used $\alpha = 0.6$ but it was not overly tuned due to computational constraints. IVON-PoCo uses outer momentum of $\rho_1 = \rho_2 = 0.3$ on CIFAR-10 and no momentum was tried on ImageNet.

### F.2. Pretraining from Scratch

We first detail the experiments described in Sec. 4.2. We use the same set-up as in (Shen et al., 2024) which follows the nanoGPT repository found under `https://github.com/karpathy/nanoGPT`. We use an effective batch size of 480 achieved via 48 gradient accumulation steps to train a 125M parameter GPT-2 model from scratch on ca. 50B tokens from OpenWebtext in 100,000 steps. The AdamW run uses a learning rate of $6 \cdot 10^{-4}$, $(\beta_1, \beta_2) = (0.9, 0.95)$. The IVON run uses a learning rate of 0.3, $(\beta_1, \beta_2) = 0.9, 0.99999$, an effective sample size $\kappa = 1 \cdot 10^{10}$, a Hessian initialization of 0.001 as well as element-wise clipping of 0.001. For IVON-PoCoMo we use the same hyperparameters but starting at step 50,000 we add a correction with a megabatch that is 10 times the size of a minibatch and megabatch statistics that get updated after every 10 inner loop steps. We also use $\rho_1 = 0.6$ and $\rho_2 = 0.1$ for momentum. All these experiments are run on 8xA100 GPUs for up to one and a half days using bf16 and flash attention (Dao et al., 2022).

For the ablations we follow a similar recipe but rather use a smaller GPT-2 model which we downsize to ca. 33M parameters by using only 4 layers and 4 heads with an embedding dimension of 512. We run the experiments on wikitext103 and use the official train-validation splits for training and evaluation. Hyperparameters for IVON, IVON-PoCo, and AdamW are kept as above. We warmup IVON-PoCo and $\alpha$-SVRG with IVON and AdamW for 5,000 steps for the ablation over $\alpha$ and for 1,000 steps for the ablation over the refresh rate of the outer estimates. We use a batch size of 64 and a short context length of 128.

### F.3. Continual Pretraining

In this experiment we continually pretrain the GPT2-125M model from Shen et al. (2024) which is publicly available (`https://huggingface.co/team-approx-bayes/gpt2-small`) which was trained for 50B tokens on OpenWebText. All methods use a context length of 512 and a batch size of 80 and are trained on a single NVIDIA A100 80GB GPU with bf16 and flash attention to speed up training and reduce GPU memory utilization. We use the first 1B tokens from the Fineweb-edu-sample-10BT which is available openly on Huggingface under the following link `https://huggingface.co/datasets/HuggingFaceFW/fineweb-edu` and split off the final 2000 documents for a validation split. For all experiments the learning rates are annealed to zero.

For IVON and IVON-PoCo, we continue using the optimizer state from pretraining but change some of the hyperparameters. We change the ess to $3 \cdot 10^{10}$ and use $\beta_2 = 0.9999$. The learning rate is set to $0.04$ instead of $0.3$, which was used for pretraining. With larger learning rates we found IVON to become less stable. For AdamW we start the optimizer state freshly and use a learning rate of $1 \cdot 10^{-4}$, down from the $6 \cdot 10^{-4}$ which was used for AdamW-trained models in Shen et al. (2024). Above, we found that loss sharply increased early in training which could lead to more forgetting of the data it was trained on. We use $\beta_1 = 0.9$ and $\beta_2 = 0.999$ which are oft-used for finetuning and the default choice in huggingface transformers (Wolf et al., 2020), which we use to implement our experiments.

For $\alpha$-SVRG and IVON-PoCo we warmstart training with 1,000 steps of AdamW and IVON, respectively, and use 40 inner steps before refreshing the outer gradient and Hessian estimates using 40 randomly sampled batches of size 80, i.e., 3200 examples and up to 1,638,400 tokens in total for estimating the gradients. We sample these batches randomly, so they need not overlap with the batches used in the following inner loop. We have found this to perform better in small experiments. Potentially, randomly sampling the batches reduces bias towards the same data used in the inner loop. For IVON-PoCo we set $\rho_1 = 0.3$ and $\rho_2 = 0.05$.

### F.4. Finetuning

We finetune various models following the Transformer architecture (Vaswani et al., 2017). First, we use Vision Transformers (Dosovitskiy et al., 2021) with ca. 88M parameters. Our experiment uses OpenCLIP (Ilharco et al., 2021) and we only train the vision encoder but not the text encoder which produces label embeddings to which the image embeddings are matched. We train for 5 epochs on Cars (Krause et al., 2013), DTD (Cimpoi et al., 2014), GTSRB (Houben et al., 2013) and RESISC45 (Cheng et al., 2017) and start correction for IVON-PoCo after just 50 steps. We use a batch size of 8, 32 warmup steps, $\beta_1 = 0.9$, $\beta_2 = 0.99999$, the Hessian is initialized to 0.1, ess equals $1 \cdot 10^{10}$, and the learning rate is initialized to 0.3 and annealed to zero.

Next, we finetune two LLMs. First, we finetune the full Qwen2.5-0.5B-Instruct model on the first 50% of the training set of XSUM (Narayan et al., 2018) and evaluate on the corresponding test split. We finetune for a single epoch with a learning rate of 0.01, $\beta_1 = 0.9$, $\beta_2 = 0.99999$, ess of $1 \cdot 10^{10}$ and a Hessian initialized to 0.001 as well as element-wise clipping to 0.001. We use $\alpha = 0.7$ and refresh the outer gradients and Hessians every 50 steps. We use the same hyperparameters for LoRA-finetuning of LLAMA-3.1-8B but increase the learning rate to 0.05. We train the model for 3 epochs on GSM8k (Cobbe et al., 2021) and calculate the loss on both input and output tokens with a standard cross-entropy criterion. For all LLM methods we use greedy decoding and zero-shot prompting.

