# OpenReview forum: "SVRG and Beyond via Posterior Correction"
_ICML.cc/2026/Conference — ICML 2026 spotlight_

### Official Review · Reviewer_Ps52 · 2026-02-23

**Soundness:** 3
**Presentation:** 4
**Significance:** 2
**Originality:** 2
**Overall Recommendation:** 5
**Confidence:** 4

**Summary:**

This paper build on top of recent Bayesian Learning Rule paper, and extends the "posterior correction" method of (Khan, 2025) to recover Bayesian SVRG-like variants of SGD (VSGD-Poco), Adam (IVON-Poco) and Newton's method (VON-Poco).

This work also provides relevant experiments that compare the performance of the novel methods introduced with their standard correspondent on logistic regression, computer vision, but also Language model and ViT benchmarks. The paper show an improvement of performance using their posterior corrected algorithms on convex problems over their standard bayesian version (eg VSGD-Poco vs VSGD in Fig 3).

**Compliance With Llm Reviewing Policy:**

Affirmed.

**Final Justification:**

See my last answer to the authors.

Score increased from 4 to 5.

**Key Questions For Authors:**

- 1) SAGA paper (Defazio et al 2014) is not cited, which is surprising in page 2
- 2) Surprising figure 8 : how is it possible to have a the beginning of the correction without a flat curve for the number of gradients calculation in the mega-batch? Fig 8 b) is the learning rate for AdamW is properly tuned?
- 3) I would emphasize that it does not seems that this work contradicts Defazio and Bottou and improve deep learning
- 4) "Boost training" is often used. It is not clear it means improve test performance or accelerate training.
- 5) Where is VSGD derived and defined?
- 6) Missing $\eta$ in the denominator.
- 7) What do the authors mean in line 206 "only batch setting"?
- 8) "The only difference is in the use of learning rate" -> a bit vague. More precisely the learning rate also acts as a dampening factor of previous $\bf{\lambda}_{in}$
- 9) Line 237 "new perspective of SVRG". How new is it? It's true that initial angle of interpretation was variance reduction of the gradient estimator. But form the original paper, the outer loop full batch gradient has been referred to as a \emph{snapshot} capturing a more consistent (because on a full batch of data) descent "knowledge" at a previous iterate
- 10) line 310 : "A posterior constructed using mega-batches is not ideal" -> vague statement. Can the authors elaborate in the paper? It feels the extension to $\alpha$-SVRG is artificial or at least arbitrary. Especially given the apparent contradiction in the way to schedule $\alpha$ from lines 323 to 329. What happened for increasing and decreasing schedules? And compared to standard SVRG with $\alpha = 1$?
- 11) Typo in the legend of Figure 3 : "SVRG improves SVRG"
- 12) Figure 4 and 5 : It's not relevant to compare the number of steps between methods having a gradient computation cost per step different.

Nitpicking
- almost same notation for inner loop size and mean vector if the posterior

**Limitations:**

yes

**Strengths And Weaknesses:**

**Strengths**
- 1) Strong mathematically motivated recover of SVRG and Adam-SVRG equivalent with a Bayesian approach leveraging Posterior Corrections
- 2) Diverse experimental sections including logistic regression, Resnet and language model pretraining (on GPT2-125M)

**Weaknesses**
- 3) The proposed method is not better than standard SVRG in Logistic regression cases, improvement not better than SGD or AdamW in Resnet experiments. Improvement with IVON or IVON-Poco vs AdamW on GPT2, but is the learning rate sweep fair? Note also that in section 4.1 using a similar constant learning rate for all methods does not seem a fair choice. One would expect to be able to use larger step sizes for SVRG than for SGD. Cf Section 4.1 of (Defazio & Bottou, 2019), which is beneficial for VR methods.
- 4) There is no ablation concerning the effect of the inner loop size $m$ and it's effect on the overall computation cost. This has been shown to be a critical additional hyperparameter cf (Defazio & Bottou, 2019) and (Sebbouh et al, 2019), that SGD and Adam(W) do not need to tune.
- 5) The section 3.4) and in general the study of the comparison of IVON-Poco and AdamW in terms of memory and computation is to shallow. In IVON's paper, it was already the case that at larger scale, memory and computation for IVON were higher than for Adam without more details. I believe section 3.4 should be extended theoretically with big $O$ notations to make clear the computational and memory costs of each step of IVON-Poco is similar to those of AdamW.

Several typos to correct. I did not check the proofs yet.

---

> ### Author Rebuttal · Authors · 2026-03-31
>
> Dear Reviewer,
>
> Thank you for your careful review. We will correct the typos and address your questions below.
>
> > “Improvement with IVON or IVON-Poco vs AdamW on GPT2, but is the learning rate sweep fair?”
>
> Yes, we used established learning rates wherever possible and performed sweeps otherwise. For example, we mention a parameter sweep in line 1003 for the results shown in Fig. 8 b). Similarly, for Fig. 4, we use learning rates that are also used in other papers like Shen et al. 2024.
>
> > “One would expect to be able to use larger step sizes for SVRG than for SGD.”
>
> Yes, this is correct. In fact, this is the idea behind our logistic regression experiments. For SGD, we show the result for the largest possible learning rate without divergence, and then we use variance reduction on this SGD to stabilize the update. If it helps, we will add such an ablation on wikitext103 to test this for LLMs.
>
> > “[N]o ablation concerning the effect of the inner loop size”
>
> We have such an ablation in Fig. 7 (right), where we fix the megabatch size but vary the inner loop size. A larger inner loop means fewer megabatch refreshes which can be cheaper but at the cost of a stale megabatch gradient that can degrade performance. Too frequent refreshes can also slightly degrade performance while being computationally less efficient.
>
> > “I believe section 3.4 should be extended theoretically with big O notations”
>
> Thank you for this suggestion, we will add a more thorough discussion of computational and memory costs.
>
> > “SAGA paper (Defazio et al 2014) is not cited”
>
> Thank you, we will add this.
>
> > “How is it possible to have a the beginning of the correction without a flat curve for the number of gradients calculation in the mega-batch?”
>
> Thanks for pointing this out. This is due to a slight mistake in plotting. We will correct this. Note that it does not change the overall form of the plot or end result.
>
> > ”I would emphasize that it does not seems that this work contradicts Defazio and Bottou”
>
> Thank you for the suggestion, we will add this point.
>
> > “"Boost training" is often used. It is not clear it means improve test performance or accelerate training.”
>
> Thank you, we will make it clearer that it means accelerating training.
>
> > “Where is VSGD derived and defined?”
>
> VSGD is SGD with Gaussian noise on the weights, similar to Khan & Rue 2023, Sec. 1.3.1. We will introduce and define it properly.
>
> > “What do the authors mean in line 206 "only batch setting"?”
>
> This is meant to read “full-batch setting”. We will correct this, thank you!
>
> > “[T]he learning rate also acts as a dampening factor of previous lambda_in.”
>
> The additional damping goes away for isotropic Gaussians because of the use of base measure. This point is a bit technical and explained in App. A; see Eq. 21 where the last term is the base measure and the theorem afterwards shows that it cancels out the damping factor.
>
> > “Line 237 "new perspective of SVRG". How new is it? It's true that initial angle of interpretation was variance reduction of the gradient estimator. But form the original paper, the outer loop full batch gradient has been referred to as a \emph{snapshot} capturing a more consistent (because on a full batch of data) descent "knowledge" at a previous iterate”
>
> Thanks for pointing this out. Could you help us find the reference where this is mentioned? We were not able to find it on our own. As per our statement in the paper, it refers to the fact that our work connects SVRG to Posterior Correction which is a unifying mechanism for knowledge transfer and adaptation. We will rephrase the sentence to make the distinction clearer and also refer to the paper you mention.
>
> > “line 310 : "A posterior constructed using mega-batches is not ideal" -> vague statement. Can the authors elaborate in the paper? It feels the extension to alpha -SVRG is artificial or at least arbitrary. Especially given the apparent contradiction in the way to schedule from lines 323 to 329. What happened for increasing and decreasing schedules? And compared to standard SVRG with alpha = 1?”
>
> By “not being ideal”, what we mean is that using megabatches, instead of full batches, deviates from the original SVRG idea and downweighting the contribution (with $\alpha$) can help mitigate this. We stress that the introduction of $\alpha$ in our framework is not arbitrary: raising posteriors to a power is similar to using learning rates in the Bayesian Learning Rule (see Eq. 19 where $\eta$ and $\alpha$ have similar roles). Where we differ is on how we use $\alpha$: unlike $\alpha$-SVRG, where $\alpha$ is always decreased, we increase it (e.g., we start with 0 and then switch it to non-zero value later). In fact, we also found that slowly increasing it from 0.2 to 0.8 further improves performance. We will add this comparison of different schedules for the wikitext103 setting:
>
> | Method  | Perplexity $\downarrow$ |
> |---|---|
> | $\alpha=1$  | 57.0  |
> | $\alpha$-SVRG  | 50.6  |
> | Ours  | 47.3  |

---

> > ### Author Rebuttal · Reviewer_Ps52 · 2026-04-02
> >
> > I thank the authors for their answers to all the points I raised. I believe the submission with the above-mentionned corrections should be accepted. I am willing to increase my score if the following concerns are fully addressed
> > - the learning rate sweeps / hyperparameters taken from previous works make the comparisons with AdamW/SGD fair (addressed, cf my other comment)
> > - the apparent contradiction between the linear decreasing schedule of $\alpha$-SVRG and the proposed increasing schedule of this submission (still lacks of an explanation or an intuition)
> > - I am still not fully convinced by the claim that suggested methods are faster than AdamW on Deep Learning tasks, as visible on Figure 8 (b), AdamW converges faster (to a higher PPL but it might be a tuning artifact) than all other methods

---

> > > ### Author Response · Authors · 2026-04-04
> > >
> > > Dear Reviewer,
> > >
> > > We are happy to see that our rebuttal was useful and also that the corrections are sufficient for our paper’s acceptance. Below is our response to your additional questions and we hope that you will increase your score. Thank you!
> > >
> > > > “the learning rate sweeps / hyperparameters taken from previous works make the comparisons with AdamW/SGD fair (addressed, cf my other comment)”
> > >
> > > Thank you for noting that this has been addressed. We will further emphasize in the paper that the hyperparameters are also used in the work by Liu et al. 2024, who perform a grid search over learning rates (as mentioned in App. C.1. of their paper).
> > >
> > > > “the apparent contradiction between the linear decreasing schedule of -SVRG and the proposed increasing schedule of this submission (still lacks of an explanation or an intuition)”
> > >
> > > One intuition comes from the Bayesian perspective (mentioned in L324-326) which implies that the outer-loop $\hat{q}_\text{out}$ becomes more informative as learning progresses. Therefore, the $\alpha$ should be increased. This is also well-supported by the second intuition of seeing variance reduction as knowledge transfer (mentioned in L21), where past knowledge becomes more useful as the model learns more and more patterns. We will expand this point further in the paper to make it more convincing.
> > >
> > >
> > > > “I am still not fully convinced by the claim that suggested methods are faster than AdamW on Deep Learning tasks, as visible on Figure 8 (b), AdamW converges faster (to a higher PPL but it might be a tuning artifact) than all other methods”
> > >
> > > Perhaps there is a confusion: in Fig. 8b, we do not  claim that our methods are faster than AdamW. We will fix the writing to avoid this confusion. What we want to say is that $\alpha$-SVRG does not improve over Adam, but IVON-PoCoMo does improve over IVON. We agree with you that Adam is in fact better than IVON-PoCoMo. We will make this clearer in the next version of the paper.
> > >
> > > ## References
> > >
> > > Liu, Hong, et al. "Sophia: A scalable stochastic second-order optimizer for language model pre-training." ICLR 2024.

---

### Official Review · Reviewer_mMdQ · 2026-03-10

**Soundness:** 4
**Presentation:** 3
**Significance:** 3
**Originality:** 3
**Overall Recommendation:** 5
**Confidence:** 4

**Summary:**

The paper derives a novel connection between stochastic variance reduction gradient (SVRG) and the Bayesian approach of posterior correction (PoCo). The connection allows showing that SVRG is a special case of PC making use of an isotropic Gaussian posterior distribution. Using a multivariate Gaussian or a Gaussian with diagonal covariance matrix instead leads to SVRG-like versions of Variational Online Newton and Improved Variational Online Newton (IVNO), respectively. The algotihms are testet for regression models and neural networks, showing that variance reduction has a positive effect for both, for neural Networks however there it does not lead to better results than Adam.

**Compliance With Llm Reviewing Policy:**

Affirmed.

**Key Questions For Authors:**

Are there any practical benefits you are expecting for neural network training?
Since variance can also have inplicit regularization effects, could reducing variance also make generalization worse?
Are there any advantages of IVNO over first order methods from your results?
Why do you use IVONPoCoMo for GPT2 and IVONPoCo for Imagenet? Does the other perfrom worse than Adam, respectively?

**Limitations:**

Yes

**Strengths And Weaknesses:**

Strength:

The theoretical conncetion between SVRG and PC is to the best of my knowledge new and very interesting.

The experimental analysis also covers larger models, like GPT2

The paper is nicely written and easy to follow.


Weakness:

Regarding the number of samples required/ the computational complexity the newly discovered variance reducing techniques do not show a real benefit in practice. Also the variance is known to have some implicit regularization effects for neutral Networks. This opens the question, if one really expects practical benefits in this direction. However, this does not mean, that the theoretically insights are not nevertheless interesting, but it may restrict their practical impact.

Comparing the regression results in Fig. 3 and Fig. 4 it seems that IVNO-PoCo does not perform better than the first order methods. Therefore, it is not clear to me if it has any advantage. Maybe you could also compare them in one figure to make comparision more easy.


Minor:
- The aberivation VSGD was not defined and the refference of the variational stochastic gradient decent (i.e. Chen et all, 25) is missing.
- VB is defined to be variational Bayes and variational Bayesian
- Page 3: Fisher -> Fisher information matrix
- Section 3.1 could rather be in the background section
- Q in line 240 should be q

---

> ### Author Rebuttal · Authors · 2026-03-31
>
> Dear Reviewer,
>
> Thank you for your review and for highlighting the novel connection, extensive experiments, and readability of our work.
> We address your questions below.
>
> > “Also the variance is known to have some implicit regularization effects for neutral Networks. This opens the question, if one really expects practical benefits in this direction”
> > “Are there any practical benefits you are expecting for neural network training? Since variance can also have inplicit regularization effects, could reducing variance also make generalization worse?”
>
> We agree that variance can have implicit-regularization effects that can, for example, help with generalization. However, we do not find strong evidence that variance-reduction interferes with this. In our experiments, we never observed a negative change in the final performance, for example, see Fig. 4 and 5 where our variance-reduced versions obtain very similar performance in the end. Noise injection in IVON might have also helped here; it is similar to the explicit regularizer used Geiping et al. 2022 where strong generalization is still observed despite non-stochastic full-batch training.
>
> Regarding expected practical benefits, we do believe that practical benefits are possible via knowledge transfer, but further investigation is needed in future works. One direction that we are pursuing is to exploit the double-loop structure to reduce computations, for instance, by curating better mega-batches and using prioritized sampling. We are currently exploring such ideas and have encouraging initial results.
>
> > “Comparing the regression results in Fig. 3 and Fig. 4 it seems that IVNO-PoCo does not perform better than the first order methods. Therefore, it is not clear to me if it has any advantage. Maybe you could also compare them in one figure to make comparision more easy.”
>
> IVON-PoCo indeed converges faster than first order methods but due to the different y-axis ranges and separate plots it is difficult to spot the faster convergence of IVON-PoCo. We will improve the plot to make this more clearly visible. Thank you for pointing this out.
>
> > “Are there any advantages of IVON over first order methods from your results?”
>
> Could you please clarify which first order method you are referring to? Is it SGD, Adam, or SVRG?
>
> > “Why do you use IVONPoCoMo for GPT2 and IVONPoCo for Imagenet?”
>
> The reason for this is that IVON-PoCoMo and IVON-PoCo give similar results and we decided to only put one of them in the plot.
>
> > The aberivation VSGD was not defined and the refference of the variational stochastic gradient decent (i.e. Chen et all, 25) is missing.
>
> Thank you for noticing this, we will properly define the abbreviation and add a reference. Please note though that our VSGD is different from Chen et al. 2025 and rather corresponds to the algorithm from Khan & Rue, Sec. 1.3.1.
>
>
> We will also correct the typos, thank you for spotting them!
>
> ## References
>
> Khan, Mohammad Emtiyaz, and Håvard Rue. "The Bayesian learning rule." Journal of Machine Learning Research 24.281 (2023): 1-46.

---

> > ### Author Rebuttal · Reviewer_mMdQ · 2026-04-03
> >
> > Thanks a lot for your answers. They clearified my main concerns. Regading which first order method I was reffering to: Any or all of them. Just was interested in general.

---

> > > ### Author Response · Authors · 2026-04-08
> > >
> > > Dear Reviewer,
> > >
> > > Thank you for your response. We are happy to hear that all of your concerns are addressed. Regarding the comparison between IVON and first-order optimizers: there is an advantage of using IVON over other first-order optimizes for convex problems. For example, compared to SVRG, IVON-PoCo converges faster for logistic regression.

---

### Official Review · Reviewer_GM97 · 2026-03-13

**Soundness:** 3
**Presentation:** 3
**Significance:** 3
**Originality:** 3
**Overall Recommendation:** 5
**Confidence:** 3

**Summary:**

This paper discusses a new connection between stochastic variance reduced gradient (SVRG) and posterior correction, by showing the former algorithm is formally reducible to a special case of the latter proposed algorithm, when specialized to an isotropic Gaussian posterior. The motivating context for SVRG is that of variance reduction, beneficial to speeding up training. Based on this connection, the authors propose extensions of SVRG motivated by other exponential family posteriors, which involve a Hessian correction and an ADAM-like modification. They perform empirical studies which suggest that particularly the latter algorithm can lead to faster training in modern deep learning architectures like LLMs.

**Compliance With Llm Reviewing Policy:**

Affirmed.

**Final Justification:**

Although the immediate benefits of the ideas in this work are not clear for deep learning settings, their clear exposition and potential for different algorithm designs are convincing enough. I am suggesting the work be accepted.

**Key Questions For Authors:**

I wonder if one possible explanation for the lack of as noticeable a speed-up in deep learning as in convex problems is that the variance of the gradient is playing a very different role in deep learning. While the rationale behind SVRG may be sound for convex, near-or-sub-quadratic problems (e.g., logistic regression), recent works on the distinct dynamics behind large-step gradient descent and the classical gradient flow seems to suggest there is yet a lot more to be understood about training in non-convex functions than was initially thought. For example, some literature has suggested that a different flow better interpolates iterates of gradient descent that exhibits "progressive sharpening" in deep learning objectives than the usual gradient flow. Phenomena such as "progressive sharpening" and "edge of stability" tend to exacerbate themselves with stochastic gradients, but quadratic functions are not rich enough to distinguish these from simple divergence of the iterates.

**Limitations:**

Yes.

**Strengths And Weaknesses:**

The strength of the work lies in the originality of the connection drawn between two distinct ideas and some concrete algorithmic proposals based on it. The motivating observation of the authors appears to be that the natural gradient is reducible to the usual one for a Gaussian distribution. Such a view is intuitively appealing from the point of view of established, optimization-based learning, as it naturally leads to algorithms with a non-negligible Hessian contribution, as the authors discuss. One limitation I can think of, which the authors fully acknowledge, is that the proposed ADAM-like modification of SVRG does not lead to as strong a speed-up as in convex problems.

---

> ### Author Rebuttal · Authors · 2026-03-31
>
> Dear Reviewer,
>
> Thank you very much for your positive review.
>
> >I wonder if one possible explanation for the lack of as noticeable a speed-up in deep learning as in convex problems is that the variance of the gradient is playing a very different role in deep learning.
>
> Yes, it is certainly possible that the variance plays a different role in neural networks, although we do not find strong evidence that it interferes with properties such as edge-of-stability and gradient flow. This is because, in our experiments, we never observed a negative change in the final performance, for example, see Fig. 4 and 5 where our variance-reduced versions obtain very similar performance in the end. Noise injection in IVON might have also helped here; it is similar to the explicit regularizer used Geiping et al. 2022 where strong generalization is still observed despite non-stochastic full-batch training.
>
> Even though variance reduction did not yield a net benefit in speed yet, it also did not hurt the speed (e.g., see Fig. 5 right panel where IVON-PoCo almost coincides with IVON). We believe that there are perhaps other ways to reduce computations, for instance, by curating better mega-batches and using prioritized sampling. We are currently exploring such avenues and have encouraging results.
>
> ## References
> Geiping, Jonas, et al. "Stochastic training is not necessary for generalization." ICLR 2022.

---

> > ### Author Rebuttal · Reviewer_GM97 · 2026-04-02
> >
> > I thank the authors for their response, which addresses my question. Although the immediate benefits of the ideas in this work are not clear for deep learning settings, their clear exposition and potential for different algorithm designs are convincing enough. I am raising my score accordingly.

---

> > > ### Author Response · Authors · 2026-04-04
> > >
> > > Dear Reviewer,
> > >
> > > Thank you for your response and raising your score, we are happy to see that your question has been addressed!

---

### Official Review · Reviewer_wz7X · 2026-03-16

**Soundness:** 3
**Presentation:** 3
**Significance:** 3
**Originality:** 4
**Overall Recommendation:** 5
**Confidence:** 3

**Summary:**

This paper establishes the connection between Stochastic Variance Reduced Gradient (SVRG) and recently proposed Bayesian method named 'posterior correction' (PoCo) for knowledge-transfer and continual learning. The main contribution is that SVRG can be recovered as a special case of PoCo when applied to isotropic Gaussian posteriors. New extensions of SVRG can be made using flexible exponential family distributions in PoCo. A Newton-like generalization, VON-PoCo, and a Adam-like generalization, IVON-PoCo, are studied for logistic regression, LLM pertaining and image classification.

**Compliance With Llm Reviewing Policy:**

Affirmed.

**Key Questions For Authors:**

1. What are the computational complexity of the new extensions? It would good to have a table listing them explicitly.

2. In the numerical experiments, please also report computing time since IVON-PoCoMo has much added computation compared with vanilla IVON.

**Limitations:**

Yes, as admitted by authors, the proposed new algorithms do not provide much speed-up in deep learning yet.

**Strengths And Weaknesses:**

# Strength

Nice connection between SVRG and posterior correction is established. New insight on gradient-corrections in SVRG is offered as knowledge transferring between old and new gradients.

# Weakness

The benefit of new extensions to deep learning is limited so far.

---

> ### Author Rebuttal · Authors · 2026-03-31
>
> Dear Reviewer,
>
> Thank you for your positive review of our work. We address your key questions below.
>
> > “What are the computational complexity of the new extensions? It would good to have a table listing them explicitly.”
>
> Thank you for the suggestion. We will add a table to include a more detailed analysis of computational complexities to show that the additional calculation is due to two additional gradients and sampling parameters. Essentially, compared to SGD, SVRG triples the amount of gradient computations which is due to the full-batch outer gradient and two inner gradients. This also triples the memory complexity. For VSGD-PoCo, we have an additional sampling step with complexity $\mathcal{O}(d)$ for a $d$-many parameter model to add noise. The same is true for Adam, alpha-SVRG, and IVON-PoCo, where alpha-SVRG is (roughly) three times more expensive, and IVON-PoCo adds an additional sampling step. We will add the exact complexity in the form of a table.
>
> > “In the numerical experiments, please also report computing time since IVON-PoCoMo has much added computation compared with vanilla IVON.”
>
> Thank you for this suggestion. We have a time comparison in Fig. 8b for continual pretraining of GPT-2, and we will add similar plots for other experiments as per your suggestion.

---

> > ### Author Rebuttal · Reviewer_wz7X · 2026-04-05
> >
> > Thanks for your clarification on the complexity. I am positive about the acceptance and please do include new results on time consumption.

---

> > > ### Author Response · Authors · 2026-04-08
> > >
> > > Dear Reviewer,
> > >
> > > Thank you for your response, we are glad that your question is clarified and will make sure to include the new results.

---

### Decision · Program_Chairs · 2026-04-30

**Decision:**

Accept (spotlight)

**Comment:**

This paper uncovers novel theoretical connections between SVRG and posterior correction and leverages this to propose extensions of SVRG based on exponential family posteriors. The reviewers agreed that the theoretical results are insightful, the experiments are well-conducted, and that the paper is clear and readable. This paper is particularly noteworthy as it connects two distinct ideas and derives new algorithms based on this. Reviewers noted (and the authors acknowledge in the paper as well) that the current practical benefit to deep learning is somewhat limited relative to convex problems. Nevertheless, this paper is likely to serve as a solid basis for future algorithmic developments. Overall, this paper is a strong, insightful contribution and I am pleased to recommend acceptance.